# THE COHERENCE TRAP: WHEN MLLM-CRAFTED NARRATIVES EXPLOIT MANIPULATED VISUAL CONTEXTS

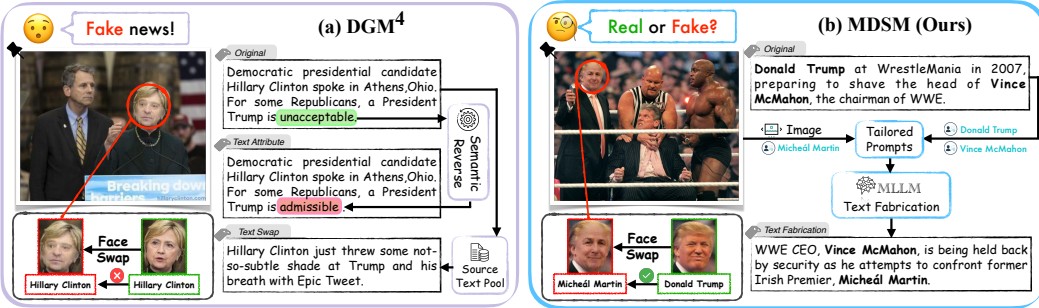

Figure 1: Comparison between widely-used DGM$^4$ (Shao et al., 2024) and MDSM (ours). (*left*) DGM$^4$ typically treats visual context manipulation as two independent procedures as rule-based text editing and image editing. The absence of context integration often results in poorly aligned samples, which can be readily perceived by the public. (*right*) In a real-world scenario, we usually face well-wrapped content on purpose. To mimic such a situation, we propose a new manipulation paradigm, which explicitly provides the modified image as well as meta info of facial editing to Multimodal Large Language Model (MLLM). Then we harness MLLM to generate contextually consistent, deceptive texts to form the challenging image-text pairs.

## ABSTRACT

The detection and grounding of multimedia manipulation has emerged as a critical challenge in combating AI-generated disinformation. While existing methods have made progress in recent years, we identify two fundamental limitations in current approaches: (1) Underestimation of MLLM-driven deception risk: prevailing techniques primarily address rule-based text manipulations, yet fail to account for sophisticated misinformation synthesized by multimodal large language models (MLLMs) that can dynamically generate semantically coherent, contextually plausible yet deceptive narratives conditioned on manipulated images; (2) Unrealistic misalignment artifacts: currently focused scenarios rely on artificially misaligned content that lacks semantic coherence, rendering them easily detectable. To address these gaps holistically, we propose a new adversarial pipeline that leverages MLLMs to generate high-risk disinformation. Our approach begins with constructing the MLLM-Driven Synthetic Multimodal (MDSM) dataset, where images are first altered using state-of-the-art editing techniques and then paired with MLLM-generated deceptive texts that maintain semantic consistency with the visual manipulations. Building upon this foundation, we present the **A**rtifact-aware **M**anipulation **D**iagnosis via MLLM (AMD) framework featuring two key innovations: Artifact Pre-perception Encoding strategy and Manipulation-Oriented Reasoning, to tame MLLMs for the MDSM problem. Comprehensive experiments validate our framework's superior generalization capabilities as a unified architecture for detecting MLLM-powered multimodal deceptions. In cross-domain testing on the MDSM dataset, AMD achieves the best average performance, with 88.18 ACC, 60.25 mAP, and 61.02 mIoU scores.

## 1 INTRODUCTION

Generative techniques have become a transformative force in artificial intelligence (Wu et al., 2024; Peng et al., 2025; Cheng et al., 2025; Fu et al., 2025; Bin et al., 2023; Zhang et al., 2025), showing remarkable adaptability across various domains and modalities. These advancements, while enriching multimedia content, also pose significant challenges to information security. In the media industry in particular, maliciously fake content manipulated by such models can profoundly mislead audiences (Zellers et al., 2019; Shao et al., 2022). The unchecked spread of fake media has already negatively affected political, financial, and other sectors (Cantarella et al., 2023; Petratos, 2021; Rocha et al., 2021), gradually becoming a major social issue (Olan et al., 2024).

While various fake news scenarios have been explored, including Luo et al. (2021) investigation of out-of-context social image-text pairs and Shao et al. (2023) work on detecting randomly tampered regions or words. Our analysis reveals two critical limitations in existing research: 1) Neglect of emerging risks from MLLMs: Current paradigms predominantly address rule-based text manipulation, overlooking the sophisticated linguistic capabilities of modern multi-modal large language models (MLLMs). MLLM-generated text exhibits superior fluency and contextual coherence, significantly increasing deception potential and public susceptibility. 2) Semantic misaligned artifacts. Most methodologies independently manipulate visual and textual elements, producing semantically discordant multimedia outputs. This misalignment not only renders manufactured disinformation too simplistic to effectively deceive the public, but also fails to replicate real-world adversarial behavior, as sophisticated attackers typically maintain meticulous visual-textual consistency to maximize manipulative impact. Both limitations render the multi-modal disinformation scenarios considered in existing works insufficiently realistic.

To address these weaknesses, we take MLLM into consideration and focus on detecting the semantic-aligned manipulation. We first construct the MLLM-Driven Synthetic Multi-modal (MDSM) dataset, which manipulates image and text in a coordinated fashion using MLLM. For the image manipulation, we consider the typical Face Swap and Face Attribute editing. For text, We innovatively guide MLLM to generate modality-aligned yet misleading fake narratives using image editing metadata. As shown in Fig. 1(b), after replacing Donald Trump's face with Micheál Martin's, we use the swapped name, Micheál Martin, to guide MLLM in generating text, ensuring that the named entity in the text aligns with the image. Following this strategy, we construct over 441k sample pairs.

The alignment of modalities and the authentic texts from MLLMs pose significant challenges for the detection of manipulated media. First, the strategy of perceiving inconsistencies between images and text through contrastive learning, as employed by prior works (Zhang et al., 2024; Shao et al., 2023), is ineffective in MDSM where images and text are well-matched already. Merely observing aligned image-text pairs is inadequate for reliable detection. Consequently, external clues and contextual knowledge are essential. Second, existing architectures like ASAP (Zhang et al., 2024) and HAMMER (Shao et al., 2023), which feature multiple detection and grounding heads, are complex and lack generalizability to unseen media sources. To address these challenges, we propose **A**rtifact-aware **M**anipulation **D**iagnosis via MLLM (AMD), which leverages MLLMs' comprehensive understanding of real-world multimedia and their ability to provide unified textual outputs. And AMD generates detection and grounding results in a coherent, text-based format, offering a more intuitive and generalized solution. In summary, we highlight our contributions of this paper as follows:

- We make an early exploration to detect and ground the MLLM-driven manipulation in multimedia and establish an MLLM-Driven Synthetic Multimodal (MDSM) dataset, which defines a more challenging and practical problem for misinformation detecting.

- We propose an Artifact-aware Manipulation Diagnosis framework (AMD) for the MDSM problem that synergizes artifact pre-perception encoding and manipulation-oriented reasoning to effectively adapt MLLMs for precise manipulation analysis.

- Comprehensive evaluations validate AMD's effectiveness and generalization capability, outperforming existing methods while maintaining parameter efficiency. With only 0.27B parameters, AMD achieves the best domain generalization average performance on both MDSM (88.18 ACC) and DGM[4] (74.47 ACC).

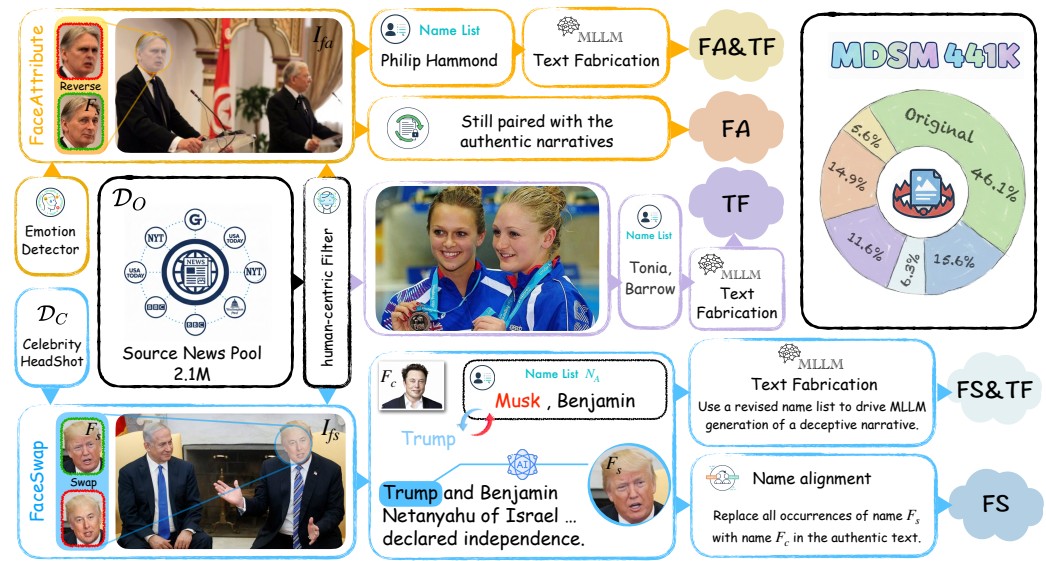

Figure 2: MDSM Construction Pipeline. □: The emotion detector helps the face editing model reverse the facial expression. The manipulated image is paired with MLLM-generated fabricated text for the *Face Attribute&Text Fabrication category (FA&TF)*, or with authentic text for the *Face Attribute category (FA)* . □: The MLLM fabricated text paired with authentic image for the *Text Fabrication category (TF)*. □: After swapping the face, the name list is updated for text-image alignment, and the manipulated image is paired with MLLM-generated fabricated text for the *Face Swap&Text Fabrication category (FS&TF)*, or with aligned authentic text for the *Face Sap category (FS)*.

## 2 MDSM DATASET CONSTRUCTION

As shown in Fig. 2, the collected source news data undergoes two key synthesis processes: 1) advanced image editing models generate visual manipulations, and 2) MLLMs produce text narratives that are semantically aligned with these visuals. We elaborate on these processes below.

### 2.1 MULTI-MODAL MEDIA SOURCE COLLECTION

We use the GoodNews (Biten et al., 2019), VisualNews (Fuxiao et al., 2020), and N24News (Zhen et al., 2021) datasets as the Source News Pool $\mathcal{D}_O$, which consists of over 2.1M image-text pairs sourced from various real-world news outlets. Given the significant influence of human-centric news among various forms of multi-modal media, we focus on human-centric data for MDSM. $\mathcal{D}_O$ is firstly filtered by detecting faces in images with Dlib (King, 2009) and identifying person names in texts with BERT (Devlin et al., 2018). Only pairs, $p_s = (I_s, T_s)$, containing both faces and named entities are used for manipulation. Additionally, we collect the *Celebrity Head-shot Dataset $\mathcal{D}_C$*, which contains about 30k pairs of head-shot images and corresponding names to facilitate the aligned manipulation for Face Swap. Details are provided in the appendix.

### 2.2 MULTI-MODAL MEDIA MANIPULATION

In the image modality, two main attacks, Face Swap (FS) and Face Attribute (FA), are employed. For the text modality, we utilize advanced MLLM to generate semantic-aligned texts for the images.

▷ **Face Swap.** Face swap is a critical tool for attackers to forge images of public figures and politicians, posing threats to societal security. We use two representative face swap methods, SimSwap (Chen et al., 2020) and e4s (Liu et al., 2023b), to perform such manipulations. We prioritize modifying larger faces to target the primary subject in the image (Fig. 3c). Given a source image $I_s$, we randomly choose one of the two methods and replace the largest face $F_s$ in $I_s$ with a face $F_c$ from $\mathcal{D}_C$, generating a manipulated face swap sample $I_{fs}$. The bounding box $y_{box} = \{x_1, y_1, x_2, y_2\}$ of

Table 1: Comparison of the proposed MDSM with existing misinformation datasets, where MM Det., Text Det., Man. Type Det., and Im. GD stand for Multi-media Detection, Text Detection, Manipulation Type Detection and Image Grounding.

| Datasets | Samples | Modality | | Tasks | | | | Semantic | MLLM |
| --- | --- | --- | --- | --- | --- | --- | --- | --- | --- |
| | | Text | Image | MM Det. | Text Det. | Man. Type Det. | Im. GD | Alignment | Inclusion |
| LIAR (Wang, 2017) | 13K | ✓ | ✗ | ✗ | ✓ | ✗ | ✗ | ✗ | ✗ |
| DFIM-HQ (Mathews et al., 2023) | 140K | ✗ | ✓ | ✗ | ✗ | ✓ | ✓ | ✗ | ✗ |
| MEIR (Sabir et al., 2018) | 139k | ✓ | ✓ | ✓ | ✓ | ✗ | ✗ | ✗ | ✗ |
| MiRAGeNews (Huang et al., 2024) | 15k | ✓ | ✓ | ✓ | ✓ | ✗ | ✗ | ✗ | ✓ |
| COSMOS (Shivangi et al., 2023) | 453k | ✓ | ✓ | ✓ | ✓ | ✗ | ✗ | ✗ | ✗ |
| DGM$^4$ (Shao et al., 2023) | 230k | ✓ | ✓ | ✓ | ✓ | ✓ | ✓ | ✗ | ✗ |
| MDSM (Ours) | 441k | ✓ | ✓ | ✓ | ✓ | ✓ | ✓ | ✓ | ✓ |

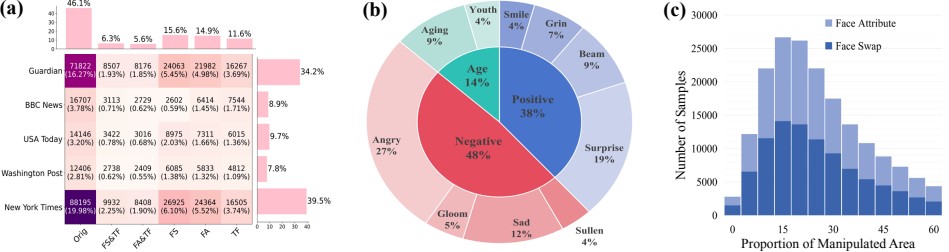

Figure 3: MDSM Statistics. (a) Distribution of media sources and manipulation categories. (b) Types of face attributes. (c) Distribution of the manipulated image area proportion of the entire image.

the swapped face and the name of $F_c$ are recorded. To keep the image-text aligned, corresponding processing is also done on the authentic text. We use MLLM to identify the name of $F_s$, as shown in Fig. 2, $F_s$ is identified as Trump. We then refine the authentic text for FS category by replacing this name with $F_c$'s name.

▷ **Face Swap and Text Fabrication.** We use Qwen2-VL (Wang et al., 2024) to generate consistent but misleading narratives. This requires knowing: 1) the inserted person's name, and 2) the names of people remaining in $I_{fs}$. Using the same strategy as above, we get $F_s$' name. We extract the full name list from the original text $T_s$ using BERT (Devlin et al., 2018), and then replace $F_s$ with inserted person's name to form the final name list $N_A$. Finally, we input $I_{fs}$ and $N_A$ into Qwen2-VL to generate aligned text.

▷ **Face Attribute.** Face emotion editing is also considered in our dataset. Our pipeline uses StyleCLIP (Patashnik et al., 2021) and HFGI (Wang et al., 2022) for attribute manipulations. Firstly, we analyze facial expressions using an emotion detector (Octavio et al., 2017) to determine positive or negative emotions. We then randomly select a method to manipulate the primary face $F_s$' attributes inversely to the classification outcome, producing $I_{fa}$. To ensure diversity, we control manipulation intensity with variable prompts and introduce age modifications. The distribution of face attribute prompts is shown in Fig. 3(b), with $y_{box}$ stored as annotation. Since the characters in $I_{fa}$ have not changed, the paired text in this category is still authentic.

▷ **Face Attribute and Text Fabrication.** Similar to face swapping, text forgery for face attribute editing is also generated by Qwen2-VL but with distinct prompts. Specifically, we instruct the MLLM to focus primarily on facial expressions to generate narratives that conform to the characters' demeanor. The input full name list is initially extracted from the source text $T_s$.

▷ **Text Fabrication.** For the TF category, we also use BERT (Devlin et al., 2018) to extract the name list $N_A$ from the original text. Then, we input $N_A$ and the original image into the MLLM to generate narratives that match the implied meaning but are still fabricated.

## 2.3 DATASET STATISTICS

With the above steps, we finally harvest our MDSM dataset $\mathcal{D}_M$, a large-scale, 100% semantic-aligned multi-modal benchmark with high-fidelity texts from MLLM. The distribution of manipulation categories is well balanced and consistent with previous datasets, ensuring fair evaluation across manipulation modes (Fig. 3a). Compared with the existing manipulation detection benchmarks in Tab. 1, MDSM has the following advantages: **1)Risk Consideration of MLLM.** MDSM acknowledges the emerging challenges posed by MLLMs and utilizes multi-modal methods to create

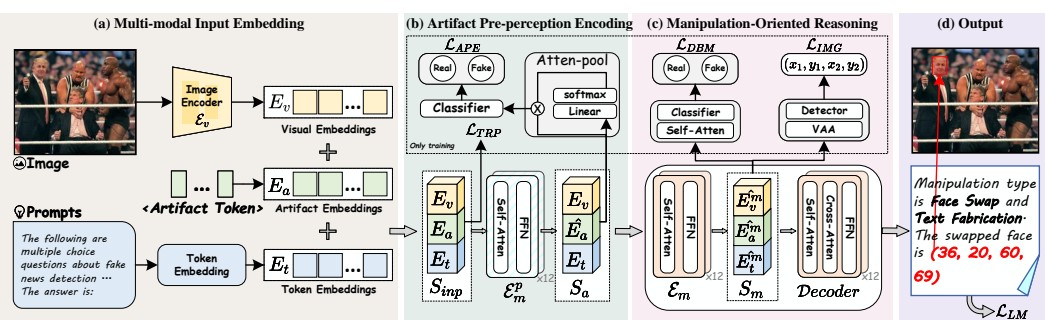

Figure 4: Overview of the proposed AMD framework. (a) Maps the manipulated image and prompts into a unified representation, incorporating an artifact token. (b) Utilizes the artifact pre-perception multimodal encoder $\mathcal{E}_m^p$ to extract perceptual clues. (c) Processes multi-modal features through $\mathcal{E}_m$ to generate text-based detection results. (d) Outputs and visualizes the final manipulation analysis.

semantically coherent and contextually plausible narratives for manipulated images. This scenario, though underexplored, is a highly significant and timely problem in the modern large model era. **2)Semantic Alignment.** MDSM is an aligned multi-modal media manipulation benchmark, which is a significant and more practical scenario for multi-modal manipulation detection. **3)Large Scale.** Our MDSM comprises 441,423 samples and is the largest benchmark for detecting and grounding multi-modal manipulation. **4)Diverse Multi-media Sources.** The multi-modal media of MDSM sources from diverse media sites, including *The Guardian*, *The New York Times*, *The Washington Post*, *USA Today*, and *the BBC*. Consequently, the generality of methods can be assessed via cross-domain evaluation.

Our proposed MDSM defines three tasks: **1)Fake Multi-modal Media Detection**. True for the manipulated media and False for the original ones. **2)Manipulation Type Detection**, recognizing Face Swap (FS), Face Attribute (FA), and Text Fabrication (TF). **3)Image Grounding**, locating the bounding box of the manipulated region in image.

## 3 METHODOLOGY

Fig. 4 illustrates the architecture of our Artifact-aware Manipulation Diagnosis framework (AMD). Built upon Florence-2 (Bin et al., 2023) to leverage real-world knowledge, AMD follows a sequence-to-sequence framework for joint textual detection and grounding. Multimodal inputs are processed through three stages, and outputs localized predictions with textual explanations.

### 3.1 MULTI-MODAL INPUT EMBEDDING.

**Prompt construction.** To adapt the MLLM for the MDSM task while preserving its inherent knowledge, we develop heuristic question(human)-answer(assistant) prompts where the image-question pair serves as input and the text response constitutes AMD's output:

*###Human:< Task >< Options >< Grounding >.*

*###Assistant:< Response >[< Coordinates >].*

In this prompt paradigm[1],

- $< Task >$: Specifies the manipulation detection objective and pairs the input image-text.
- $< Options >$: Lists all candidate answers for MDSM task.
- $< Grounding >$: Conditionally triggers region localization (via coordinates in brackets) only when image manipulation is detected.
- $< Response >$: Encapsulates the correct answers.
- $[< Coordinates >]$: Optionally encloses tampered region coordinates if the image is altered.

---

[1]Details and examples are given in supplementary materials.

**Artifact Token Embeddings.** To effectively adapt the MLLM into MDSM context while preserving its pretrained knowledge, we introduce a learnable Artifact Token that explicitly encodes artifacts from heterogeneous inputs. Formally, let the artifact token embeddings be denoted as $E_a \in \mathbb{R}^{n_a \times d}$, where $n_a$ indicates the token count and $d$ the embedding dimension. The textual input is processed through an embedding layer to obtain text embeddings $E_t \in \mathbb{R}^{n_t \times d}$, while the visual input is encoded via a vision backbone $\mathcal{E}_v$ followed by a LayerNorm-augmented linear projection, yielding image embeddings $E_v \in \mathbb{R}^{n_v \times d}$. The above embeddings are concatenated to construct the input sequence: $S_{\text{inp}} = [E_v; E_a; E_t]$, where $[\cdot; \cdot]$ means concatenating along the token dimension.

## 3.2 ARTIFACT PRE-PERCEPTION ENCODING

This stage aims to perceive manipulation artifacts within input data and condense these forensic clues into the artifact token. Specifically, the input sequence $S$ undergoes processing through the artifact pre-perception multimodal encoder $\mathcal{E}_m^p$, yielding $\hat{S} = [\hat{E}_v; \hat{E}_a; \hat{E}_t]$. To inject artifact-aware clues into the artifact token embedding $\hat{E}_a$, we pick $\hat{E}_a$ from $\hat{S}$ and feed it into an artifact-aware classification head. As illustrated in Fig. 4, this classification head is optimized via a manipulation detection objective to explicitly encode artifact-related patterns into $\hat{E}_a$.

Particularly, the embedding $\hat{E}_a$ is encoded into a global representation via weighted pooling. Firstly, the token scores $\mathcal{W} \in \mathbb{R}^{1 \times n_a}$ are calculated as:

$$\mathcal{W} = m^\top \text{ReLU}(\mathcal{M}\hat{E}_a^\top + b), \tag{1}$$

where $\mathcal{M} \in \mathbb{R}^{h \times d}$, $m \in \mathbb{R}^h$, and $b \in \mathbb{R}^h$, with $h$ as the hidden dimension. After normalizing $\mathcal{W}$ via softmax, the artifact representation $\bar{E}_a$ is derived as a weighted sum, $\text{softmax}(\mathcal{W}) \cdot \hat{E}_a$.

Then we equip a binary classifier $C_a$ to determine whether artifact traces are present:

$$\mathcal{L}_{APE} = \mathbb{E}_{(I,T) \sim \mathcal{D}_M} \textbf{CE}(C_a(\bar{E}_a), y_{fd}), \tag{2}$$

where **CE** means cross-entropy loss, $y_{\text{fd}}$ is the label of fake multimodal media detection task.

**Task Adaption & Knowledge Preservation.** To effectively inject the artifact clues into $\hat{E}_a$ without distorting the original real-world knowledge of MLLM, two strategies are adopted, we 1) freeze the parameters of $\mathcal{E}_m^p$ during artifact perception loss optimization (Eq. 2), such that allowing more artifact clues can be accumulated into the artifact token as well as preserving the raw MLLM knowledge; 2) replace the text and image embeddings in $\hat{S}$ with the original ones to preserve the original MLLM knowledge, *i.e.,* feeding $S_a = [E_v; \hat{E}_a; E_t]$ to the subsequent modules, as shown in Fig. 4.

## 3.3 MANIPULATION-ORIENTED REASONING

Manipulation-Oriented Reasoning (MOR) is in charge of generating the textual answer in response to the question prompt. To acquire an accurate response, we augment the network optimization in MOR with two guiding tasks: visual Artifact Capture via Grounding and Manipulation-focused Guidance.

**Visual Artifact Capture via Grounding.** The sequence $S_a$ is fed into multimodal encoder $\mathcal{E}_m$, resulting in a new sequence $S_m = [\hat{E}_v^m; \hat{E}_a^m; \hat{E}_t^m]$. Given that visual embeddings contain rich local spatial information related to artifact traces, we propose a *Visual Artifact Aggregation* (VAA) module to aggregate spatial information in $\hat{E}_v^m$ to perform manipulation bbox grounding. Firstly, the $\hat{E}_a^m$ is transformed into a query token $q_a \in \mathbb{R}^{1 \times d}$ using the attention-based weighted pooling (Eq. 1). Then, $q_a$ collects visual manipulation clues from image features $\hat{E}_v^m$ via cross attention:

$$u_{agg} = \text{Attention}(q_a, \hat{E}_v^m, \hat{E}_v^m). \tag{3}$$

Subsequently, the $u_{agg}$ is sent to the bbox detector to generate artifact coordinates. We follow Rezatofighi et al. (2019) to construct the image manipulation grounding loss using L1 loss $\mathcal{L}_1$ and GIoU loss $\mathcal{L}_{IoU}$:

$$\mathcal{L}_{IMG} = \mathbb{E}_{(I,T) \sim \mathcal{D}_M}(\mathcal{L}_1 + \mathcal{L}_{IoU}). \tag{4}$$

**Manipulation-focused Guidance** further highlight whether the multimodal input is manipulated or not, tuning the MLLM to be sufficiently sensitive to the fake multi-modal media. To fully capture

manipulation-related information embedded within different modalities, we propose a Dual-Branch Manipulation guidance strategy. Specifically, each modality feature in the encoder output sequence $S_m$ is treated as a query $Q$ and undergoes interaction for binary classification. Given that artifact traces predominantly appear in the image modality, the sequence composed of $\hat{E}_a^m$ and $\hat{E}_v^m$ is regarded as the image modality feature. The interaction process is formulated as:

$$u_v = \text{Attention}(\hat{E}_{v+a}^m, \hat{E}_t^m, \hat{E}_t^m), \quad u_t = \text{Attention}(\hat{E}_t^m, \hat{E}_{v+a}^m, \hat{E}_{v+a}^m), \tag{5}$$

where $\hat{E}_{v+a}^m$ represents the concatenation of $\hat{E}_v^m$ and $\hat{E}_a^m$, while $\hat{E}_t^m$ corresponds to the textual sequence. The cross-modal interaction outputs, $u_v$ and $u_t$, are respectively processed by a binary classifier $C_m$ to distinguish between manipulated and original multimodal media. Thus the Dual-Branch Manipulation guidance loss can be calculated as:

$$\mathcal{L}_{DBM} = \mathbb{E}_{(I,T)\sim \mathcal{D}_M} \sum_{x \in \{v,t\}} \textbf{CE}(C_m(u_x), y_{fd}). \tag{6}$$

**Language modeling.** The input sequence $S_a$ is processed through an encoder-decoder architecture, ultimately generating a pure text output that includes choices and coordinates (Fig. 4d) as specified in the prompts. In this stage, an autoregressive approach is adopted, where the decoder generates the target sequence $y$ conditioned on $S_m$. The language modeling loss $\mathcal{L}_{LM}$ (Bin et al., 2023) is used to supervise the decoded text outputs.

### 3.4 TOKEN REDUNDANCY PENALTY

To suppress the redundancy and increase the information density among tokens in $E_a$, we design a Token Redundancy Penalty (TRP) optimization term. Specifically, we first encourage the columns of $E_a$ to be as orthogonal as possible by introducing a loss term $\mathcal{L}_{\text{orth}}$, which increases the matrix rank. We construct Gram matrix of $E_a$, $G = E_a E_a^\top \in \mathbb{R}^{n_a \times n_a}$, and the orthogonality of the columns can be measured by the off-diagonal elements of the Gram matrix. Ideally, if the columns are orthogonal, the off-diagonal entries of $G$ should be zero. Therefore, we define:

$$\mathcal{L}_{\text{orth}} = \|G - \text{Diag}(\text{diag}(G))\|_F^2, \tag{7}$$

where $\text{Diag}(G)$ denotes a diagonal matrix retaining only the diagonal elements of $G$, and $\|\cdot\|_F$ denotes the Frobenius norm used to aggregate the differentiable loss.

To avoid a potential checkerboard pattern in $E_a$ under the constraint of $\mathcal{L}_{\text{orth}}$—which could lead to loss of information—we further introduce a modulation constraint $\mathcal{L}_{\text{mod}}$ based on the Kullback–Leibler (KL) divergence. Particularly, we first normalize the components to form a distribution: $p_{t,i} = \frac{E_{a\,t,i}^2}{\sum_{i=1}^d E_{a\,t,i}^2}$. While the target distribution is set as the even distribution ($\frac{1}{d}$), thereby encouraging each component to contain information evenly with following constrain:

$$\mathcal{L}_{\text{mod}} = \frac{1}{n_a} \sum_{t=1}^{n_a} \left( \sum_{i=1}^d p_{t,i} \log p_{t,i} + \log d \right), \tag{8}$$

Finally, the overall Token Redundancy Penalty is defined as the combination of both terms:

$$\mathcal{L}_{TRP} = \mathcal{L}_{\text{orth}} + \mathcal{L}_{\text{mod}}, \tag{9}$$

$\mathcal{L}_{TRP}$ is imposed on the $S_{inp}$ sequence during the APE stage.

### 3.5 TRAINING AND INFERENCE

**Training.** All guiding losses above and the language modeling loss are incorporated into the training process, forming a unified optimization framework as follows:

$$\mathcal{L} = \mathcal{L}_{APE} + \mathcal{L}_{DBM} + \mathcal{L}_{IMG} + \mathcal{L}_{TRP} + \mathcal{L}_{LM}, \tag{10}$$

**Inference.** All auxiliary heads for $\mathcal{L}_{APE}, \mathcal{L}_{DBM}, \mathcal{L}_{IMG}$, and $\mathcal{L}_{TRP}$ are discarded during inference. For a piece of multimodal media, the image and the question (text & prompts) follow the same steps shown in Fig. 4 and generate the textual detection and grounding results.

Table 2: Comparison of multi-modal learning methods on MDSM, where the background gray indicates the intra-domain performance. The better results in each group are in **bold**. AVG refers to the average performance across five news domains.

| Setting | Method | Test Domain | | | | | | | | | | | | | | | | | |
|---|---|---|---|---|---|---|---|---|---|---|---|---|---|---|---|---|---|---|
| | | NYT | | | Guardian | | | USA | | | Wash. | | | BBC | | | AVG | | |
| | | ACC | mAP | mIoU | ACC | mAP | mIoU | ACC | mAP | mIoU | ACC | mAP | mIoU | ACC | mAP | mIoU | ACC | mAP | mIoU |
| Zero-Shot | Qwen2.5-VL (Bai et al., 2025) | 47.74 | 29.24 | 0.00 | 35.18 | 25.70 | 0.00 | 24.66 | **40.60** | 0.00 | 25.11 | 40.29 | 0.28 | 35.89 | 31.51 | 0.00 | 33.72 | 33.47 | 0.06 |
| | Qwen3-a22b (Yang et al., 2025) | 45.29 | 25.01 | 0.71 | 38.12 | 27.41 | 1.19 | 22.87 | 39.17 | 0.10 | 22.87 | **40.77** | 1.34 | 37.22 | **31.60** | 0.98 | 33.27 | **33.69** | 0.86 |
| | GPT-4o (Hurst et al., 2024) | 48.48 | 27.90 | 0.82 | 35.68 | **29.49** | **1.23** | 24.62 | 39.88 | 1.37 | 23.62 | 38.89 | 1.22 | 37.19 | 30.48 | 1.20 | 33.92 | 33.33 | 1.17 |
| | Gemini-2.0 (Team et al., 2023) | **56.05** | **33.16** | **1.44** | **41.26** | 24.37 | 1.12 | **29.60** | 38.29 | **1.40** | **29.15** | 35.20 | **2.42** | **38.12** | 29.13 | **2.25** | **38.83** | 32.03 | **1.72** |
| Tr. on NYT | ViLT (Kim et al., 2021) | 83.27 | 64.27 | 22.73 | 72.18 | 31.76 | 20.21 | 70.34 | 36.45 | 21.48 | 65.71 | 36.23 | 17.56 | 74.33 | 36.10 | 19.36 | 73.17 | 40.96 | 20.27 |
| | HAMMER (Shao et al., 2023) | 79.20 | 55.86 | 51.34 | 68.23 | 40.10 | 21.56 | 71.52 | 41.17 | 13.74 | 68.50 | 41.47 | 13.92 | 67.37 | 42.23 | 16.12 | 70.96 | 44.16 | 23.34 |
| | HAMMER++ (Shao et al., 2024) | 79.61 | 57.06 | 54.44 | 66.99 | 38.07 | 17.34 | 67.18 | 37.58 | 10.76 | 66.28 | 37.97 | 10.88 | 66.12 | 37.82 | 13.68 | 69.23 | 41.70 | 21.42 |
| | FKA-Owl (Liu et al., 2024) | **94.67** | 78.18 | 55.81 | 77.20 | 46.88 | 43.67 | 78.00 | 44.45 | 50.73 | 75.49 | 50.83 | 43.53 | 84.65 | **60.73** | 43.28 | 81.60 | 56.77 | 46.23 |
| | AMD(Ours) | 92.24 | **84.47** | **72.94** | **80.21** | **64.00** | **62.51** | **78.56** | **68.49** | **55.17** | **82.64** | **69.41** | **56.66** | **86.14** | 60.58 | **70.54** | **83.96** | **69.39** | **63.56** |
| Tr. on Guardian | ViLT (Kim et al., 2021) | 68.80 | 43.99 | 21.77 | 85.29 | 67.34 | 41.80 | 70.34 | 46.24 | 37.68 | 78.61 | 47.17 | 38.13 | 80.00 | 44.79 | 38.97 | 76.61 | 49.90 | 35.67 |
| | HAMMER (Shao et al., 2023) | 61.89 | 37.98 | 18.84 | 78.50 | 52.40 | 51.53 | 74.78 | 50.76 | 43.40 | 75.11 | 50.34 | 46.36 | 81.32 | 50.15 | 56.03 | 74.32 | 48.33 | 43.23 |
| | HAMMER++ (Shao et al., 2024) | 62.75 | 36.45 | 23.76 | 80.95 | 59.92 | 64.67 | 75.36 | 48.77 | 47.13 | 76.30 | 49.56 | 48.91 | 80.12 | 50.36 | 57.97 | 75.10 | 49.01 | 48.49 |
| | FKA-Owl (Liu et al., 2024) | 80.60 | 40.44 | 26.33 | **92.60** | 78.24 | 71.04 | 80.90 | 51.80 | 50.93 | 78.88 | 51.62 | 50.88 | 87.61 | **68.57** | 61.24 | 84.12 | 58.13 | 52.20 |
| | AMD (Ours) | **84.29** | **48.54** | **52.38** | 91.43 | **80.85** | **85.09** | **88.80** | **53.05** | **52.51** | **86.64** | **54.07** | **53.27** | **89.74** | 64.75 | **61.82** | **88.18** | **60.25** | **61.02** |

Table 3: Comparison of multi-modal learning methods on DGM$^4$, where the guardian domain with background gray is intra-domain. $P_{tok}$ is Precision of fake token grounding.

| Method | Test Domain | | | | | | | | | | | | | | | | | | |
|---|---|---|---|---|---|---|---|---|---|---|---|---|---|---|---|---|---|---|---|
| | Guardian | | | | USA | | | | Wash. | | | | BBC | | | | AVG | | | |
| | ACC | mAP | $P_{tok}$ | mIoU | ACC | mAP | $P_{tok}$ | mIoU | ACC | mAP | $P_{tok}$ | mIoU | ACC | mAP | $P_{tok}$ | mIoU | ACC | mAP | $P_{tok}$ | mIoU |
| ViLT (Kim et al., 2021) | 68.27 | 42.29 | 69.87 | 43.19 | 52.79 | 31.28 | 62.11 | 33.78 | 55.76 | 33.26 | 57.17 | 31.10 | 44.14 | 39.68 | 59.06 | 21.96 | 55.24 | 36.63 | 62.05 | 32.49 |
| HAMMER (Shao et al., 2023) | 78.34 | 66.79 | 78.27 | 61.09 | 64.97 | 40.49 | 73.76 | 40.51 | 63.54 | 40.26 | 76.13 | 38.53 | 54.97 | 40.84 | 81.48 | 43.74 | 65.45 | 47.10 | 77.41 | 45.97 |
| HAMMER++ (Shao et al., 2024) | 79.13 | 67.11 | 78.24 | 62.15 | 65.25 | 40.74 | 73.24 | 41.14 | 63.83 | 40.34 | 76.17 | 38.21 | 54.24 | 41.25 | 81.73 | 43.23 | 65.61 | 47.36 | 77.34 | 46.19 |
| FKA-Owl (Liu et al., 2024) | 82.97 | 53.86 | **87.70** | 65.69 | 67.57 | 38.97 | **79.44** | 32.57 | 67.05 | 37.70 | **81.55** | 31.86 | 70.26 | 40.20 | **84.54** | 46.48 | 71.96 | 42.68 | **83.31** | 44.15 |
| AMD (Ours) | **84.61** | **68.50** | 82.78 | **81.24** | **70.62** | **43.20** | 75.73 | **41.99** | **70.28** | **43.36** | 77.76 | **39.05** | **72.37** | **56.57** | 83.76 | 45.20 | **74.47** | **52.91** | 80.01 | **51.87** |

# 4 EXPERIMENT

Please refer to the appendix for the experimental setup and evaluation metrics.

## 4.1 QUANTITATIVE RESULTS

**Effectiveness & Generalization.** We assess AMD against four SOTA methods on the MDSM and DGM$^4$ datasets. For MDSM (Tab. 2), we train on *The Guardian* and *NYT*, testing on the rest. For DGM$^4$ (Tab. 3), we train on the largest subset, *The Guardian*. Tab. 2 also shows zero-shot results for general-purpose models. Our key findings are: **(1) MLLMs' knowledge boosts performance.** Forgery-trace methods like ViLT (Kim et al., 2021) and HAMMER series (Shao et al., 2024) show limited performance, unlike MLLM-based methods like FKA-Owl (Liu et al., 2024) and AMD. For instance, trained on MDSM-NYT (Tab. 2), AMD achieves an 84.47 intra-domain mAP and >60 cross-domain, while HAMMER scores 57.06 and <42, respectively. **(2) AMD achieves strong grounding.** AMD attains the best average mIoU of 63.56 (NYT-trained) and 61.02 (Guardian-trained) (Tab. 2). General-purpose models perform poorly (mIoU < 3). AMD's superiority stems from its question-answer heuristic prompts and MOR module, which omits coordinate outputs when no manipulation is detected, thus reducing unnecessary errors. **(3) AMD generalizes effectively.** On DGM$^4$ (Tab. 3), AMD outperforms the HAMMER series on all metrics (74.47 ACC, 52.91 mAP, 80.01 $P_{tok}$, 51.87 mIoU). It also surpasses FKA-Owl in ACC, mAP, and mIoU, despite a lower $P_{tok}$.

**Generalization Assessment across MLLMs.** To assess generalization on different MLLMs, we evaluated an NYT-trained AMD on test narratives generated by four MLLMs: Qwen-VL (Bai et al., 2023), X-InstructBLIP (Panagopoulou et al., 2024), LLaVA (Liu et al., 2023a), and mPLUG-Owl (Ye et al., 2025). Results (Tab. 4b, chart I) show robust performance, with intra-domain (NYT) and cross-domain APs exceeding 76 and 53, respectively.

**Details of Manipulation Type Detection.** Using AMD trained on the NYT domain as an example, the bar chart II in Tab. 4b shows that text-modal (TF) manipulations are harder to detect than image-modal ones. FA achieves intra-domain AP of 88.45 and cross-domain AP of 71.37, while TF reaches 79.84 and 57.53, respectively. This highlights the deceptive nature of MLLM-generated narratives.

Table 4: Ablation on (a) each components and (b) discussion regarding performance on test set of difference MLLMs & differnece manipulation type.

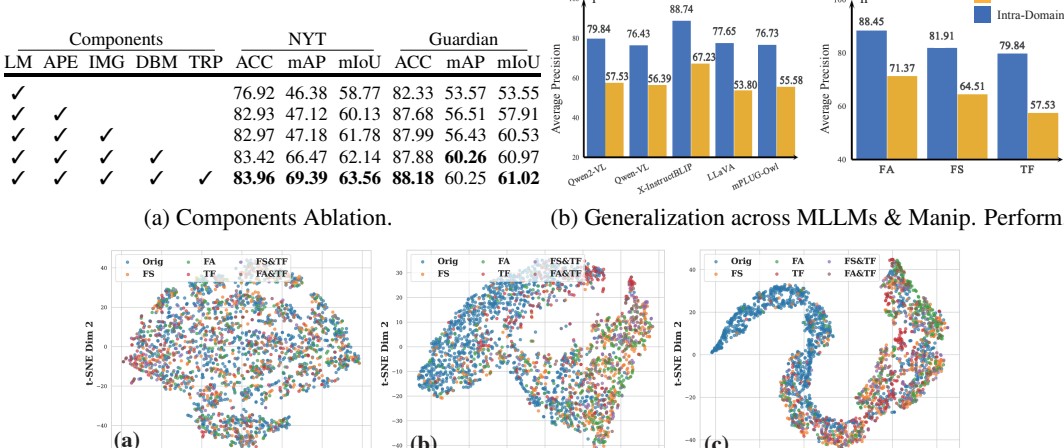

| Components | | | | | NYT | | | Guardian | | |
|---|---|---|---|---|---|---|---|---|---|---|
| LM | APE | IMG | DBM | TRP | ACC | mAP | mIoU | ACC | mAP | mIoU |
| ✓ | | | | | 76.92 | 46.38 | 58.77 | 82.33 | 53.57 | 53.55 |
| ✓ | ✓ | | | | 82.93 | 47.12 | 60.13 | 87.68 | 56.51 | 57.91 |
| ✓ | ✓ | ✓ | | | 82.97 | 47.18 | 61.78 | 87.99 | 56.43 | 60.53 |
| ✓ | ✓ | ✓ | ✓ | | 83.42 | 66.47 | 62.14 | 87.88 | **60.26** | 60.97 |
| ✓ | ✓ | ✓ | ✓ | ✓ | **83.96** | **69.39** | **63.56** | **88.18** | 60.25 | **61.02** |

(a) Components Ablation.  (b) Generalization across MLLMs & Manip. Perform.

Figure 5: Visualization of Artifact Token sequence. (a) Visualization of $E_a$ in $S_{inp}$. (b) Visualization of $\hat{E}_a$ in $S_a$. (c) Visualization of $\hat{E}_a^m$ in $S_m$

## 4.2 ABLATION STUDY

**Component Ablation.** Tab. 4a presents the results for each component considered in our study. We use a fine-tuned Florence-2 with our designed prompts as the baseline. As shown, incorporating Artifact Pre-perception Encoding (APE) improves all three task metrics, especially binary classification accuracy, which increases from 76.92 to 82.93 on NYT and from 82.33 to 87.68 on Guardian. This demonstrates that pre-perception of manipulation traces is vital for aiding MLLMs in multi-media manipulation detection. Adding auxiliary tasks, such as Dual-Branch Manipulation (DBM) and Image Manipulation Grounding (IMG), enhances fake news classification and grounding performance, while also slightly improving binary classification. Notably, DBM significantly boosts AMD's mAP, increasing from 47.18 to 66.47 on NYT and from 56.43 to 60.26 on Guardian. Furthermore, the incorporation of the Token Redundancy Penalty (TRP) yields comprehensive performance gains, especially exhibiting stable improvements in ACC and mIoU across both domains.

**Artifact Token Visualization.** Fig. 5 visualizes the Artifact token at different stages via t-SNE (van der Maaten & Hinton, 2008). As shown in Fig. 5a to c, the sample points progressively form more distinct clusters, clearly demonstrating the effectiveness of our AMD optimization in enhancing the Artifact Token's ability to distinguish between different categories.

For a more complete understanding of the MDSM dataset and the ablations on the AMD design, **we refer readers to the appendix.**

## 5 CONCLUSION

This study discloses two critical limitations in current multimedia manipulation detection: underestimation of dynamic semantic deception risks posed by MLLMs and the unrealistic, semantically incoherent misalignment artifacts among existing benchmarks. To address these challenges, we construct the MLLM-Driven Synthetic Multimodal (MDSM) dataset and the Artifact-aware Manipulation Diagnosis (AMD) framework to address this new and challenging problem. AMD integrates Artifact Pre-perception Encoding and Manipulation-Oriented Reasoning to enhance detection of MLLM-generated multimodal disinformation. Comprehensive experiments demonstrate the framework's superior generalization capabilities, validating its effectiveness as a unified solution for combating advanced MLLM-driven deception.

**Broader impact and limitation.** Our MDSM dataset simulates real-world attacks and aids in fake news detection, focusing on high-risk face manipulations in multimodal news, with scene-level manipulation reserved for future work. Our method shows promising zero-shot performance on scene-level manipulations (see appendix), with further exploration planned.

## ETHICS STATEMENT

We hereby solemnly declare that **we have carefully read the ICLR Code of Ethics, and that this research strictly adheres to these guidelines**. The MDSM dataset and associated analyses were created solely to support research on detection and mitigation of modern MLLM-driven multimodal misinformation. We recognize that assembling realistic, semantically coherent synthetic examples entails dual-use risks: the same materials and procedures could be misused to produce deceptive content. To minimize harm, we adopt a harm-minimizing, controlled-release approach: we will not publish the generation pipeline, detailed prompts, or prompt–response pairs to prevent their exploitation by adversaries for generating harmful content; public distribution is limited to vetted, research-only access under a signed Data Usage Agreement (DUA); distributed images will carry conspicuous visual watermarks and standardized metadata tags; high-fidelity originals and sensitive metadata will be withheld; images of minors and clearly sensitive contemporary conflict content have been excluded; and reserve the right to revoke access on evidence of misuse. Full technical and procedural details of these safeguards are documented in the Appendix and in the dataset README.

## REPRODUCIBILITY STATEMENT

We are committed to ensuring the reproducibility of this work. To ensure the reproducibility of our research, we provide comprehensive code, data, and experimental details. (1) **Code**: The source code for all models, along with scripts to run experiments and analyze results, will be available at github. The repository will include a detailed README.md file with instructions for installing dependencies and reproducing our findings. (2) **Data**: The dataset used in this study will be made available to researchers under a Data Use Agreement. (3) **Experimental Details**: Appendix F lists the hyperparameters for all experiments, the software and hardware environments, and the exact computation methods for our evaluation metrics.

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

APPENDIX

APPENDIX CONTENTS

## A RELATED WORK

**Deepfake Detection.** With the rapid advancement of generative models and the explosive growth of generated data, Deepfake detection technology has also progressed swiftly, enabling it to address the information risks posed by counterfeit data. Historically, Deepfake detection has been categorized into unimodal and multimodal detection approaches. Unimodal detection includes both image-based (Zhao et al., 2021; Li et al., 2020) and text-based (Sheng et al., 2022; Huang et al., 2023) methods, all of which have achieved strong intra-domain performance. Recently, with the rise of Multimodal Large Language Models (MLLMs), multimodal Deepfake detection has garnered increasing attention (Liu et al., 2025a; Shao et al., 2023; Liu et al., 2025b). Among them, HAMMER (Shao et al., 2023) combines contrastive learning to build a robust detection model that not only classifies manipulation types but also grounds manipulation locations. However, HAMMER does not address the enhancement of cross-domain performance. FKA-Owl (Liu et al., 2024), which is based on MLLM, incorporates more world knowledge to improve the model's cross-domain performance. Despite this, FKA-Owl lacks grounding functionality and, due to the incorporation of MLLM, has become particularly cumbersome.

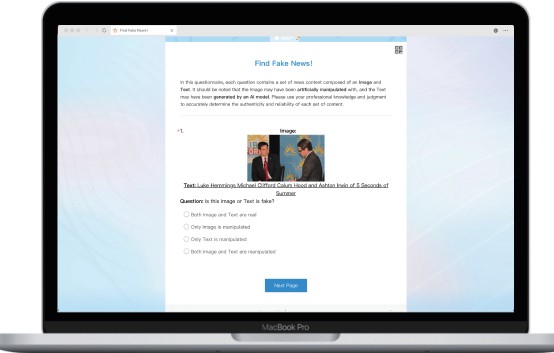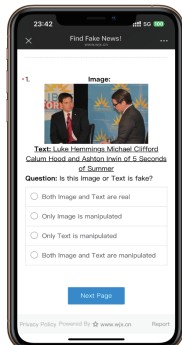

Figure 6: The user interface of the human evaluation study where each participant is given pairs of news images and caption and asked to determine whether they are manipulated or not.

**Multi-Modal Large Language Model.** In recent years, Multi-Modal Large Language Models (MLLMs) have emerged as a crucial technology for understanding and reasoning across multiple modalities, particularly text and images. By extending the capabilities of Large Language Models (LLMs) to incorporate visual inputs, these models have demonstrated outstanding performance in tasks such as image captioning and visual question answering. CLIP (Radford et al., 2021) and ALIGN (Jia et al., 2021) leveraged contrastive learning to align visual and textual representations, enabling efficient zero-shot vision-language understanding. Subsequently, models such as Flamingo (Alayrac et al., 2022) and BLIP-2 (Li et al., 2023) have introduced vision-language transformers, integrating pre-trained LLMs with vision encoders to enhance cross-modal reasoning and generative capabilities. More recently, GPT-4V (OpenAI, 2023) and Florence-2 (Bin et al., 2023) have significantly enhanced the potential of MLLMs in tackling complex multi-modal tasks by leveraging a more efficient framework and larger-scale pre-training data. A key advantage of MLLMs is their ability to acquire extensive world knowledge during large-scale pretraining, which significantly enhances their reasoning capabilities in downstream tasks.

In this work, we fully leverage the intrinsic world knowledge of MLLMs to enhance the robustness of multi-modal manipulation detection and improve inference capabilities in unknown scenarios.

## B  ASSESSING PUBLIC RISKS OF MLLM NARRATIVES - A HUMAN EVALUATION

To assess human ability to identify multimodal misinformation generated by MLLMs in combination with image manipulation models, we design a human evaluation study based on 100 image-text pairs sampled from our MDSM dataset. As illustrated in Fig. 6, each test sample belonged to one of four categories: both image and text are original (**Orig**), only image is manipulated (Fake Image, **F-I**), only text is manipulated (Fake Text, **F-T**), and both image and text are manipulated (Fake Image & Text, **F-I&T**).

We recruit 15 adult volunteers, all holding at least a bachelor's degree, to participate in the evaluation. As summarized in Fig. 7a, the accuracy of identifying Orig, F-I, F-T, and F-I&T samples was 53.19%, 13.22%, 20.18%, and 22.42%, respectively. Notably, all manipulated categories exhibit low recognition accuracy, with none exceeding 23%. To better visualize classification performance and patterns of confusion, we construct a row-normalized confusion matrix (Fig. 7b). The matrix reveals that among all manipulated types, F-I&T samples were most frequently misclassified as original real news, with a false-negative rate of 46.43%. This finding suggests that fake news jointly generated by MLLMs and image editing models can achieve high semantic coherence and pose a significant threat in terms of deception.

Considering real-world scenarios, however, humans often do not need to identify which modality is manipulated; it is often sufficient to detect the presence of any form of misinformation to avoid being misled. Therefore, we introduce two binary metrics to quantify this ability: **Overall Recall** and **False Alarm Rate**. The Overall Recall is defined as the proportion of manipulated items (F-I,

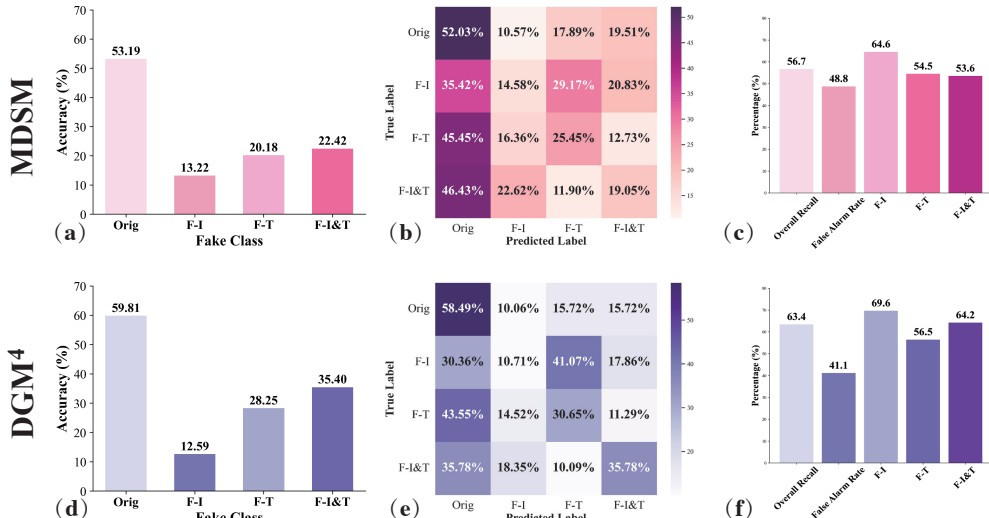

Figure 7: Human evaluation statistics on multimodal fake news identification. (a) Per-class accuracy across four types of image-text pairs on MDSM. (b) Row-normalized confusion matrix showing classification tendencies on MDSM. (c) Human perception of manipulated news and misclassification of real content on MDSM. (d) Per-class accuracy across four types of image-text pairs on DGM$^4$. (e) Row-normalized confusion matrix showing classification tendencies on DGM$^4$. (f) Human perception of manipulated news and misclassification of real content on DGM$^4$.

F-T, F-I&T) correctly identified as any type of fake (i.e., not labeled as Orig). The False Alarm Rate refers to the proportion of original items (Orig) incorrectly identified as any form of fake. We further compute the per-category fake detection rate for F-I, F-T, and F-I&T individually. The results are illustrated in Fig. 7c. The analysis shows that participants identify 56.7% of the manipulated samples as fake under this relaxed criterion. However, they also falsely flag 48.8% of the original news items as fake. This elevated false alarm rate indicates a conservative judgment tendency in the testing environment, implying that the actual detection rate in real-world conditions may be significantly lower than 56.7%. Among manipulated categories, F-I samples has the highest detection rate at 64.5%, compared to 54.5% for F-T and 53.6 for F-I&T. This discrepancy suggests that MLLM-generated textual fabrications in our MDSM dataset are particularly deceptive and challenging to identify.

Following the same evaluation setup, we also conducted an investigation on the DGM$^4$ dataset. The results, shown in Fig. 7d-f, indicate that the DGM$^4$ dataset exhibits similarly strong deception. The recognition accuracy for all manipulation samples does not exceed 36% (Fig. 7d). For the samples involving text manipulation in DGM$^4$, the recognition accuracies for F-I and F-I&T are 28.25% and 35.40%, respectively, both higher than the 20.18% and 22.42% for MDSM. This suggests that the fabricated text generated by MLLMs, as considered in MDSM, is more likely to mislead the general public.

In summary, the findings highlight a generally low human sensitivity to misinformation generated by MLLMs and image editing systems, especially in cases where multimodal manipulations are semantically consistent. **This underscores the real-world threat posed by MLLM-involved fake news and points to the urgent need for robust automatic misinformation detection systems to mitigate societal harm and support informed decision-making.**

## C  DISTRIBUTION OF MLLM-GENERATED TEXTS

To evaluate the quality of fake corpus generated by MLLM, we compare their textual distributions against the authentic news corpus in MDSM. We conduct a statistical analysis using SpaCy (ExplosionAI, 2023) and TextBlob (Loria, 2023) across five linguistic dimensions: (1) *average sentence length* (syntactic complexity), (2) *top-10 frequent words* (topical and lexical overlap), (3) *noun–verb*

Table 5: Statistical comparison results between real and MLLM-generated text.

| Metric | Authentic Corpus | MLLM-Generated Corpus |
| --- | --- | --- |
| Average Sentence Length | 20.79 | 21.98 |
| Top 10 Frequent Words | left, said, **new**, **president**,**mr**, last, center, **one**, right, **join** | **new**, **join**, us, event, **president**, **one**, seen, hosting, **mr**, york |
| Noun–Verb Ratio | 2.20 | 2.00 |
| Type–Token Ratio | 0.15 | 0.13 |
| Average Sentiment Score | 0.04 | 0.08 |

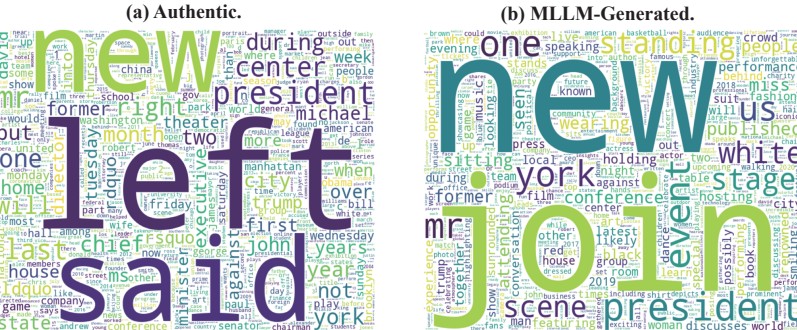

**(a) Authentic.**    **(b) MLLM-Generated.**

Figure 8: Word Clouds of Authentic and MLLM-Generated Corpus.

*ratio* (stylistic tendencies), (4) *type–token ratio* (lexical diversity), and (5) *average sentiment score* (tonal neutrality).

As shown in Tab. 5, the distributions of generated texts exhibit strong proximity to real news. The average sentence length and noun–verb ratio are nearly identical, indicating that the MLLM outputs capture comparable syntactic rhythm and stylistic balance. Frequent word distributions overlap substantially (e.g., 'new', 'president', 'mr', 'one', 'join'), as further illustrated by the word clouds in Fig. 8, reflecting clear topical alignment. The type–token ratio of generated texts is 0.13, only slightly lower than authentic news's 0.15, which is expected given structured prompts and reduced lexical randomness. Sentiment scores in both corpora remain close to neutral and slightly positive (the authentic is 0.04 and the MLLM-genarated is 0.08), consistent with the stylistic norms of mainstream journalism.

Overall, these results confirm that MLLMs, when guided by carefully designed instructions, can produce texts that closely mimic the linguistic distribution of authentic news across multiple dimensions.

# D  DISCUSSION OF MDSM AND DGM[4]

## D.1  CONTRIBUTION OF MDSM

The core contribution of our MDSM dataset is threefold: (1) It addresses the under-explored yet critical threat of misinformation crafted by MLLMs; (2) It provides the first dedicated benchmark for detecting semantically-aligned news manipulation – a significantly more realistic and challenging scenario; (3) MDSM enhances richness and scale (almost 2 times larger than DGM[4]) by collecting broader media types, news topics, and diverse sources (5 domains across 3 sources: NYT, BBC, USA Today, Guardian, Wash.) compared to prior datasets like DGM[4] (4 domains, 1 source).

Crucially, MLLM-drafted narratives are highly deceptive, as evidenced by human evaluators achieving only 22.42% accuracy (Fig. 7). This starkly contrasts with the misaligned image-text contexts in datasets like DGM[4], where human detection is far easier (35.40% accuracy). This gap underscores the unique challenge MDSM addresses.

Table 6: Discussion experiments on MDSM and DGM$^4$

| Len. | MDSM$_{Gs}$ | | | DGM$^4_G$ | | |
|---|---|---|---|---|---|---|
| | ACC | mAP | mIoU | ACC | mAP | mIoU |
| NYT | 72.26 | 40.28 | 44.70 | \ | \ | \ |
| Guardian | 87.07 | 78.32 | 80.26 | 84.61 | 68.50 | 81.24 |
| USA | 73.32 | 40.18 | 43.07 | 70.62 | 43.20 | 41.99 |
| Wash. | 70.16 | 41.17 | 42.17 | 70.28 | 43.36 | 39.05 |
| BBC | 74.63 | 50.64 | 46.93 | 72.37 | 56.57 | 45.20 |
| **AVG** | **75.49** | **50.12** | **51.43** | **74.47** | **52.91** | **51.87** |

(a) Comparation on MDSM$_{Gs}$ and DGM$^4_G$.

| Setting | HAMMER | | | AMD | | |
|---|---|---|---|---|---|---|
| | ACC | mAP | mIoU | ACC | mAP | mIoU |
| DGM$^4$ | 93.06 | 85.19 | 75.86 | 90.23 | 83.17 | 76.19 |
| MDSM$_r$ | 91.27 | 62.79 | 61.24 | 91.83 | 81.96 | 78.63 |
| **Performance Gap** | **-1.79** | **-22.40** | **-14.62** | **+1.60** | **-1.21** | **+2.44** |

(b) HAMMER and AMD's performance. Performance Gap is from DGM$^4$ to MDSM$_r$

| Methods | Test Dataset | | | | | AVG$_{ACC}$ |
|---|---|---|---|---|---|---|
| | MDSM | | | COSMOS (Shivangi et al., 2023) | MiRAGe (Huang et al., 2024) | |
| | ACC | mAP | mIoU | ACC | ACC | |
| ViLT | 39.62 | 21.18 | 21.93 | 51.27 | 31.18 | 40.69 |
| HAMMER | 46.02 | 24.88 | 35.19 | 57.14 | 34.12 | 45.76 |
| HAMMER++ | 46.17 | 24.78 | 33.78 | **57.79** | 34.56 | 46.17 |
| FKA-Owl | 54.23 | **32.48** | 36.76 | 57.02 | 38.76 | 50.00 |
| **AMD (Ours)** | **54.75** | 31.49 | **43.68** | 57.16 | **39.68** | **50.53** |

(c) Zero-Shot transfer performance of DGM$^4$-trained models on other benchmarks. AVG$_{ACC}$ is the average accuracy across all datasets in each row.

Therefore, MDSM establishes a realistic benchmark specifically designed for the emerging threat of MLLM-generated disinformation, driving essential progress in cross-modal forgery detection. With the boom of MLLMs, we need to consider the threat of their malicious use in social multimodal news to deceive the public. From this aspect, our MDSM is a timely contribution to promote this direction of research.

## D.2 CHALLENGE LEVEL OF MDSM AND DGM$^4$

It is important to clarify that the motivation behind constructing MDSM is not to create a dataset more challenging than DGM$^4$. Rather, MDSM addresses a fundamentally distinct threat: MLLM-crafted, semantically aligned multimodal disinformation, which remains an underexplored vulnerability in existing benchmarks. This represents a critical and emerging risk paradigm that necessitates dedicated investigation.

To enable a fair comparison, we reduce the MDSM-Guardian subset (MDSM$_{Gs}$) to match the 103k samples of the DGM$^4$-Guardian subset (DGM$^4_G$). Next, we train AMD on this subset and present the results in Tab. 6a. The MDSM$_{Gs}$-trained AMD achieves 75.49 ACC / 50.12 mAP / 51.43 mIoU, which are comparable to the results on DGM$^4$ (74.47 / 52.91 / 51.87, respectively). These comparable results suggest that MDSM and DGM$^4$ present similar levels of challenge under controlled conditions.

Additionally, we reduce the size of MDSM to match that of DGM$^4$, denoting this subset as MDSM$_r$, and conduct an evaluation without domain shift. As shown in Tab. 6b, AMD demonstrates strong performance on both datasets, with the performance gap within ±3. However, HAMMER exhibits substantial performance degradation on MDSM$_r$, particularly in terms of mAP (22.40 points drop) and mIoU (14.62 points drop). These results indicate that existing frameworks struggle to effectively handle semantically aligned manipulations.

The above experimental results indicate that MDSM and DGM$^4$ have similar levels of challenge, and MDSM is slightly more difficult in distinguishing manipulation types at a finer granularity. Additionally, our human verification results confirm this (the human discrimination accuracy rates for MDSM and DGM$^4$ are 22.42% and 35.69%, respectively).

## D.3 CROSS EVALUATION AMONG MDSM AND DGM$^4$

The evaluation of our MDSM-trained models on DGM$^4$ and other benchmarks is reported in Tab. 10, where AMD achieves the best overall performance. We further train the models on DGM$^4$ and test them on MDSM and other benchmarks. As shown in Tab. 6c, AMD consistently demonstrates

strong generalization to unseen cases, achieving the highest performance with an MDSM-ACC of 54.75, MDSM-mIoU of 43.68, and MiRAGE-ACC of 39.68, while also attaining the best average ACC performance of 50.53. Notably, on the MiRAGE dataset, all MDSM-trained models (except ViLT) achieve an accuracy above 50 (Tab. 10), whereas DGM$^4$-trained models consistently fall below 40. These results highlight the significant potential of MDSM-trained models in detecting purely generated multimodal disinformation.

# E   PROMPT PARADIGM

## E.1   PROMPT FOR AMD

The details of the heuristic question-answer prompts in AMD are as follows:

### *###Human:*

```
The following are multiple choice questions about fake news
detection.  The text caption of the news is:  <Text>.  The
identity and emotion of the face, and the semantic and sentiment
of the text should not be manipulated.  Question:  Is there any
face swap/attribute or text fabrication in the news?

A. No.

B. Only face swap.

C. Only face attribute.

D. Only text swap.

E. Both face swap and text fabrication.

F. Both face attribute and text fabrication.

If there is manipulation of a face, locate the one most likely
manipulated face in the image and append the results to your
selected option.

The answer is:
```

### *###Assistant:*

```
<Option>[Manipulated face:  <loc_x1><loc_y1><loc_x2><loc_y2>]
```

Where $<Text>$ refers to the textual narratives paired with the input image, $<Option>$ represents the correct answer option for this sample, such as *E. Both face swap and text fabrication.* And $<loc\_>$ is added to the vocabulary as a special token representing coordinates. Fig. 9 shows two kinds of prompts.

## E.2   PROMPT FOR GENERAL-PURPOSE MODEL

To ensure fairer testing and more credible results for general-purpose models in Tab.2, we enhanced the invocation of general-purpose models by adding more detailed descriptions to the AMD prompt, as follows:

### *###Human:*

```
<Same as AMD>

If face manipulation, use rectangular box coordinates in the
format of [x1,y1,x2,y2], where the top-left vertex of the image
is defined as (0,0) and the bottom-right vertex as (1,1) for
relative positioning, and append the results to the option you
have selected.

DO NOT output analysis.  ONLY output final answer in format:
[Option + Coordinates (if applicable).]
```

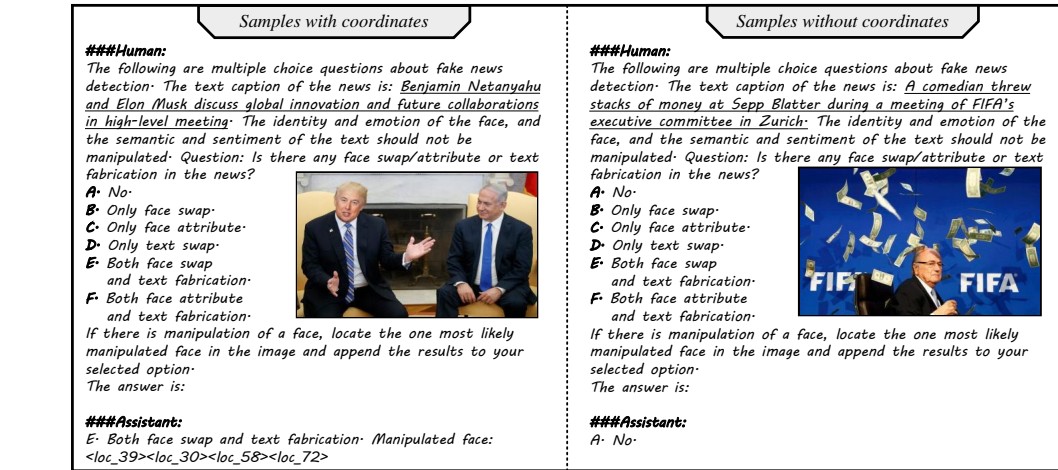

Figure 9: Examples of Image-Prompt pairs in AMD.

# F EXPERIMENTAL SETUP

## F.1 IMPLEMENTATION DETAILS

All experiments are conducted on 4 NVIDIA GeForce RTX 4090 GPUs using **Distributed Data Parallel (DDP)** training in PyTorch. The image encoder $\mathcal{E}_v$ is based on DaViT (Ding et al., 2022), with Florence-2-B (Bin et al., 2023) serving as the backbone. The APE $\mathcal{E}_m^p$ is based on the Florence-2 encoder and remains frozen during training. The classifiers and bounding box (bbox) detector consist of two Multi-Layer Perceptron layers, with output dimensions of 2 and 4, respectively. For manipulation detection guidance, the **Dual-Branch Manipulation** shares a common classifier.

The training images are resized to $224 \times 224$ and undergo random horizontal flipping. The batch size per GPU (**per_GPU_bs**) is set to 6, and the model is trained for 12 epochs. We use the AdamW optimizer (Loshchilov & Hutter, 2017) with an initial learning rate of $1e^{-7}$ and a weight decay of 0.01. A cosine learning rate scheduler with a warm-up phase is applied, gradually increasing the learning rate to $1e^{-6}$ in the first 1000 steps, and then decaying it to $1e^{-7}$ throughout training. Our code will be released to provide further implementation details.

## F.2 BASELINES

We adapt four state-of-the-art multi-modal methods to the MDSM setting for comparison, including three multi-modal manipulation detection models and one multi-modal learning approach:

- **HAMMER** (Shao et al., 2023) is a pioneering model for the multi-modal manipulation detection and grounding. It employs two unimodal encoders to extract visual and textual forgery features, which are then aligned through contrastive learning. Following this, a multi-branch transformer architecture with two specialized decoders is utilized for manipulation detection and grounding.

- **HAMMER++** (Shao et al., 2024) is a more powerful model that builds upon HAMMER by integrating contrastive learning from both global and local perspectives.

- **FKA-Owl** (Liu et al., 2024) is another pioneering model designed for large vision-language models to perform multi-modal fake news detection, and it demonstrates outstanding cross-domain performance. Since FKA-Owl does not support fine-grained classification tasks, we fine-tuned it using the same prompts as those used for AMD.

- **ViLT** (Kim et al., 2021), for the multi-modal learning approach, is a representative single-stream method where cross-modal interaction layers operate on the concatenation of image and text inputs. For adaptation, We add classification and detection heads to the corresponding outputs of the model.

Table 7: Ablation of artifact token length (a), knowledge preservation strategies (b), TRP position(c), and discussion of inference speed (d).

| Len. | NYT | | | Guardian | | |
|---|---|---|---|---|---|---|
| | ACC | mAP | mIoU | ACC | mAP | mIoU |
| 16 | 83.24 | 63.40 | 57.46 | 83.16 | **60.92** | 59.86 |
| 32 | **83.96** | **69.39** | **63.56** | **88.18** | 60.25 | **61.02** |
| 64 | 72.23 | 48.20 | 51.26 | 83.53 | 47.18 | 51.55 |

(a) Artifact Token Length.

| Method | NYT | | | Guardian | | |
|---|---|---|---|---|---|---|
| | ACC | mAP | mIoU | ACC | mAP | mIoU |
| **AMD** | **83.96** | **69.39** | **63.56** | **88.18** | **60.25** | **61.02** |
| w/o Replace | 75.08 | 56.46 | 53.19 | 82.60 | 50.93 | 51.22 |
| w/o Frozen | 77.21 | 50.62 | 54.16 | 82.98 | 54.10 | 50.79 |

(b) Knowledge Preservation Strategies.

| Position | NYT | | | Guardian | | |
|---|---|---|---|---|---|---|
| | ACC | mAP | mIoU | ACC | mAP | mIoU |
| $S_{inp}$ | **83.96** | 69.39 | **63.56** | **88.18** | **60.25** | **61.02** |
| $S_a$ | 82.83 | **69.47** | 62.17 | 88.03 | 60.15 | 60.81 |
| $S_m$ | 55.97 | 48.27 | 49.41 | 64.52 | 45.15 | 51.53 |

(c) TRP Position.

| Method | Params (M) | | Throughput (p/s) | |
|---|---|---|---|---|
| | Total | Trainable | Train | Inference |
| ViLT | 121.07 | 121.07 | 1.85 | 2.38 |
| HAMMER(++) | 441.12 | 228.25 | 28.97 | 61.28 |
| FKA-Owl | 6771.98 | 33.55 | 1.25 | 1.33 |
| **AMD** | **276.95** | **276.95** | **5.55** | **13.38** |

(d) Inference Efficiency Comparison.

## F.3 EVALUATION METRICS

To comprehensively evaluate our proposed MDSM, we follow the rigorous evaluation protocols and metrics outlined in Shao et al. (2023) for all manipulation detection and grounding tasks. The detailed evaluation setup is organized as follows:

- **Binary Classification. Accuracy (ACC)** is adopted as the evaluation metric to measure the correctness of real/fake news classification results.
- **Multi-Label Classification.** For multi-label classification tasks, we employ **mean Average Precision (mAP)**. This metric effectively captures the average performance across all labels, providing a comprehensive assessment of multi-dimensional manipulation type classification accuracy.
- **Manipulated Image Bounding Box Grounding.** To evaluate the precision of predicted manipulated bounding boxes, we calculate the **mean Intersection over Union (mIoU)** between the ground-truth and predicted coordinates for all testing samples. This metric quantifies the spatial overlap between detected regions and actual manipulated areas, reflecting the localization accuracy of the model.
- **Manipulated Text Token Grounding.** In the DGM$^4$ benchmark, an additional task of manipulated text token grounding is included. For this task, **Precision** is used as the evaluation metric to measure the accuracy of identifying manipulated text tokens within input sequences.

This standardized evaluation framework ensures a systematic and comparative assessment of MDSM across diverse manipulation scenarios, aligning with both general detection tasks and benchmark-specific requirements.

## G ABLATION STUDY

**Impact of Artifact Token Length.** We examine the effect of Artifact Token length on performance in APE. As shown in Tab. 7a, an Artifact Token length of 32 yields the best results. Specifically, in the NYT domain, AMD with 32 Artifact Tokens achieves the highest scores: 83.96 ACC, 69.39 mAP, and 63.56 mIoU.

**Efficacy of Knowledge Preservation Strategies.** We compare the efficacy of two knowledge preservation strategies in APE: freeze $\mathcal{E}_m^p$ and replace embeddings, as shown in Tab. 7b. The results indicate that without these strategies, all three metrics experience a decline, with the most significant drop in grounding performance. For instance, in the NYT domain, omitting the frozen and replace strategies reduces mIoU from 63.56 to 53.19 and 54.16, respectively.

**Impact of TRP Position.** We examine the effect of applying TRP at different AMD stages, as shown in Tab. 7c. Introducing TRP to $S_{\text{inp}}$ and $S_a$ in the APE stage improves performance, with $S_{\text{inp}}$

Table 8: Modality ablation of AMD in MDSM, where the background gray indicates the intra-domain performance. The better results in each group are in boldface. AVG refers to the average performance across the five news domains.

| Train | Modality | Test Domain | | | | | | | | | | | |
|---|---|---|---|---|---|---|---|---|---|---|---|---|---|
| | | NYT | | Guardian | | USA | | Wash. | | BBC | | AVG | |
| | | ACC | mIoU | ACC | mIoU | ACC | mIoU | ACC | mIoU | ACC | mIoU | ACC | mIoU |
| NYT | MDSM-Image | 87.03 | 69.20 | 77.65 | 62.09 | 75.20 | 53.27 | 78.71 | 54.99 | 81.11 | 69.96 | 80.14 | 61.90 |
| | MDSM-Text | 80.10 | - | 72.99 | - | 73.10 | - | 68.91 | - | 72.97 | - | 73.61 | - |
| | **MDSM** | **92.24** | **72.94** | **80.21** | **62.51** | **78.56** | **55.17** | **82.64** | **56.66** | **86.14** | **70.54** | **83.96** | **63.56** |
| Guardian | MDSM-Image | 78.76 | 50.09 | 86.11 | 84.33 | 81.58 | 50.10 | 81.07 | 52.12 | 82.10 | 59.20 | 81.92 | 59.17 |
| | MDSM-Text | 73.29 | - | 84.18 | - | 70.16 | - | 71.22 | - | 74.10 | - | 74.59 | - |
| | **MDSM** | **84.29** | **52.38** | **91.43** | **85.09** | **88.80** | **52.51** | **86.64** | **53.27** | **89.74** | **61.82** | **88.18** | **61.02** |

producing better results; for example, 88.18 ACC, 60.2 mAP and 61.02 mIoU in the Guardian domain, all surpassing $S_a$. In contrast, applying TRP to $S_m$ in the MOR stage causes a sharp performance drop, likely because $S_m$ has already captured task-relevant information, and TRP forces harmful information loss.

**Efficiency Discussion.** Tab. 7d compares params scale and throughput (images-text pairs per second) on RTX 4090. With 276M parameters, AMD is substantially smaller than FKA-Owl (6771M), enabling faster training and inference. Among comparable-sized models like ViLT and HAMMER, AMD achieves slower speed than HAMMER but significantly outperforms them on MDSM tasks. Overall, AMD delivers strong cross-domain performance while maintaining a compact architecture and efficient inference.

**Modalities Ablation.** To validate the significance of multi-modal correlation in our proposed AMD framework, we isolate the inputs that correspond solely to the image modality (MDSM-Image) or the text modality (MDSM-Text). The results in Tab. 8, indicate that the lack of modalities has a certain impact on AMD's performance. However, AMD still manages to achieve relatively robust results. For instance, in the MDSM-Image scenarios of the two training domains, the average ACC is over 80, and the decrease in the average mIoU is no more than 3 percentage points. When only the text modality is kept, the ACC performance drops notably, with the average ACC being approximately 73. This is partially in line with the results of the manipulation type detection precision discussed in Section 4.1, where it was found that text modality manipulation is harder to detect than image manipulation. This can be a crucial direction for optimizing future forgery detection models.

# H    EVALUATION OF FINE-TUNED LARGE MLLMS

To further evaluate the capability of general-purpose LLM-based VLMs on MLLM-involved fake multimedia, we fine-tuned and tested the Qwen2.5-VL-3B and -7B (Bai et al., 2025) on MDSM. Fine-tuning was performed with LoRA (r = 16, $\alpha$ = 16) using a learning rate of $2 \times 10^{-5}$ for two epochs; results are reported in Tab. 9. After two epochs, both Qwen2.5-VL variants achieve competitive in-domain accuracy (e.g., on the NYT split Qwen2.5-VL-7B reaches 72.00 ACC while Qwen2.5-VL-3B reaches 69.29), which is substantially higher than the zero-shot ACC, 33.72, reported in Table 2. However, despite its relatively small size ($\approx$ 0.3B parameters), our AMD achieves the best overall performance across domains. Notably, larger models exhibit stronger out-of-domain generalization: Qwen2.5-VL-7B attains 72.53 ACC in the Guardian domain and maintains more than 60 ACC in several other domains, including 73.21 ACC on the Wash. All in all, these results indicate that while in-domain fine-tuning improves performance for large VLMs, purpose-built models such as AMD remain highly effective and more competitive in the multi-domain setting.

Table 9: Comparison of fine-tuned Qwen2.5-VL for MDSM, where the background gray indicates the intra-domain performance. The better results in each group are in boldface. AVG refers to the average performance across the five news domains.

| Train Domain | Method | NYT | | | Guardian | | | USA | | | Wash. | | | BBC | | | AVG | | |
|---|---|---|---|---|---|---|---|---|---|---|---|---|---|---|---|---|---|---|---|
| | | ACC | mAP | mIoU | ACC | mAP | mIoU | ACC | mAP | mIoU | ACC | mAP | mIoU | ACC | mAP | mIoU | ACC | mAP | mIoU |
| NYT | Qwen2.5-VL-3B (Bai et al., 2025) | 82.42 | 57.20 | 47.78 | 66.40 | 45.25 | 37.71 | 64.04 | 58.72 | 49.61 | 63.06 | 45.82 | 39.80 | 70.52 | 59.65 | 35.67 | 69.29 | 53.33 | 42.11 |
| | Qwen2.5-VL-7B (Bai et al., 2025) | 83.49 | 58.36 | 61.36 | 72.63 | 45.22 | 40.56 | 66.64 | 58.45 | **56.07** | 66.15 | 49.32 | 36.39 | 71.09 | **64.64** | 56.22 | 72.00 | 55.20 | 50.12 |
| | AMD(ours) | **92.24** | **84.47** | **72.94** | **80.21** | **64.00** | **62.51** | **78.56** | **68.49** | 55.17 | **82.64** | **69.41** | **56.66** | **86.14** | 60.58 | **70.54** | **83.96** | **69.39** | **63.56** |
| Guardian | Qwen2.5-VL-3B (Bai et al., 2025) | 63.29 | 44.89 | 46.29 | 72.48 | 63.59 | 51.18 | 70.96 | 50.78 | 55.73 | 72.93 | 57.31 | 59.67 | 59.50 | 55.90 | 55.92 | 67.83 | 54.49 | 53.76 |
| | Qwen2.5-VL-7B (Bai et al., 2025) | 60.14 | 42.59 | **55.01** | 72.53 | 67.35 | 56.31 | 70.63 | 51.78 | **56.99** | 73.21 | **58.87** | **60.28** | 67.04 | 58.90 | **64.65** | 68.71 | 55.90 | 58.65 |
| | AMD(ours) | **84.29** | **48.54** | 52.38 | **91.43** | **80.85** | **85.09** | **88.80** | **53.05** | 52.51 | **86.64** | 54.07 | 53.27 | **89.74** | **64.75** | 61.82 | **88.18** | **60.25** | **61.02** |

Table 10: Zero-Shot transfer performance of MDSM-trained models on different benchmarks. $\text{AVG}_{\text{ACC}}$ is the average accuracy across all datasets in each row.

| Method | DGM[4] (Shao et al., 2023) | | | COSMOS (Shivangi et al., 2023) | MiRAGe (Huang et al., 2024) | $\text{AVG}_{\text{ACC}}$ |
|---|---|---|---|---|---|---|
| | ACC | mAP | mIoU | ACC | ACC | |
| ViLT | 39.10 | 22.77 | 32.11 | 39.76 | 39.13 | 39.33 |
| HAMMER | 49.22 | 28.22 | 44.31 | 52.12 | 53.18 | 51.51 |
| HAMMER++ | 50.51 | 29.19 | 46.77 | 52.60 | **53.92** | 52.34 |
| FKA-Owl | 56.18 | **33.71** | 22.10 | **53.78** | 52.20 | 54.05 |
| **AMD(Ours)** | **56.52** | 31.02 | **47.07** | 52.48 | 53.23 | **54.08** |

# I  ZERO-SHOT DETECTION AND GROUNDING

To evaluate the generalization capability of the models on unseen data and their cross-dataset adaptability, we conduct zero-shot testing of ViLT (Kim et al., 2021), the HAMMER series(Shao et al., 2024), FKA-Owl(Liu et al., 2024), and AMD models. They are trained on the MDSM dataset and evaluated across the following three public datasets:

- **DGM**[4] (Shao et al., 2023): Focuses on complex image-text multimodal manipulation scenarios, supporting both manipulation detection and grounding.
- **COSMOS** (Shivangi et al., 2023): Targets scenarios involving text replacement leading to image-text inconsistency, and supports binary classification of image-text pairs as paired or not.
- **MiRAGe** (Huang et al., 2024): Characterized by fully generated images, and supports binary classification of image-text pairs as real or fake.

The zero-shot results are presented in Tab. 10. Our AMD model achieves the best results on DGM[4], with an ACC of 56.52 and an mIoU of 47.07, outperforming other comparative models. Its mAP of 31.02 ranks second only to FKA-Owl's 33.71. On COSMOS, AMD achieves results comparable to those of HAMMER and FKA-Owl. Notably, the image-text pairs in COSMOS are composed of mismatched real texts and real images without any traces of tampering or manipulation, which may explain AMD's limited performance on this dataset. In the fully generated MiRAGe dataset, AMD obtains an ACC of 53.23, second only to HAMMER++'s 53.92. AMD achieves the highest average ACC performance of 54.08. This indicates that AMD also has significant potential in detecting purely generated fake image-text data.

# J  GENERALIZATION ACROSS DIFFUSION-BASED SYNTHETIC CONDITIONS

Considering that in real-world scenarios fake news images may be generated through diverse manipulation paradigms, we design extensive experiments to assess the generalization ability of MDSM-trained

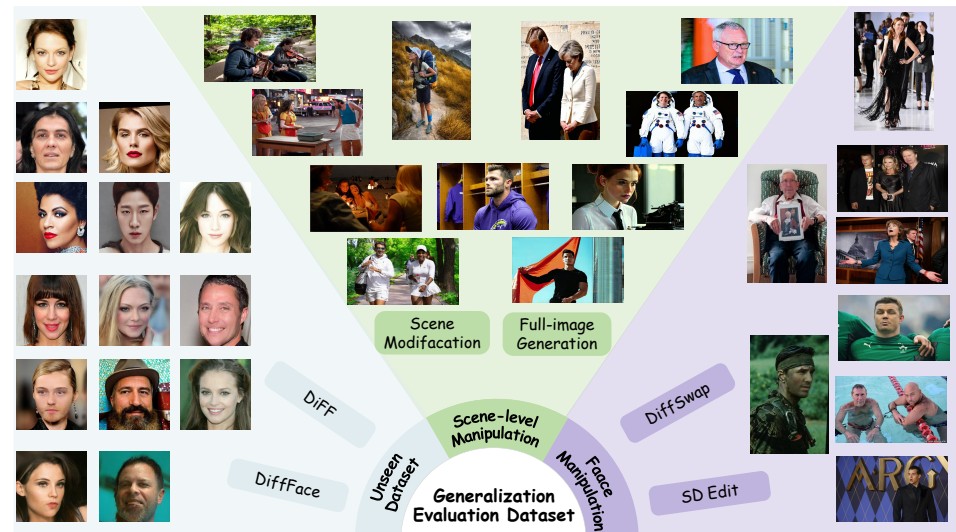

Figure 10: Examples of Generalization Evaluation Dataset.

Table 11: ACC performance of MDSM-trained models on DiFF and DiffFace. The better results in each group are in boldface. AVG refers to the average accuracy across the test dataset.

| Method | Test Dataset | | | | | | | |
| --- | --- | --- | --- | --- | --- | --- | --- | --- |
| | DiFF | | | | DiffFace | | | |
| | FaceSwap | FaceEdit | Image2Image | AVG | DiffSwap | Text2Image | DDIM | AVG |
| HAMMER (Shao et al., 2023) | 71.21 | 52.35 | 91.09 | 71.55 | 50.73 | 55.31 | 89.07 | 65.04 |
| HAMMER++ (Shao et al., 2024) | **71.98** | 52.79 | 91.23 | 72.00 | 51.23 | 58.15 | 91.65 | 67.01 |
| ViLT (Kim et al., 2021) | 64.23 | 51.78 | 79.24 | 65.08 | 46.15 | 51.28 | 76.18 | 57.87 |
| FKA-Owl (Liu et al., 2024) | 69.25 | 53.79 | **99.13** | **74.06** | 51.02 | 59.77 | 96.23 | 69.01 |
| **AMD(Ours)** | 68.62 | **54.80** | 98.20 | 73.87 | **51.41** | **59.80** | **98.38** | **69.86** |

models when encountering unseen manipulations. All test data, as shown in Fig. 10, are generated by Stable Diffusion–based models that are not included in MDSM. We evaluate ViLT, the HAMMER family, FKA-Owl, and our proposed AMD, all trained on the MDSM-Guardian domain.

**Unseen Dataset.** We conduct evaluations of the models on two additional synthetic face benchmarks built upon Stable Diffusion, namely DiffFace (Kim et al., 2022) and DiFF (Cheng et al., 2024). These datasets include diffusion-based face-swap manipulations (FaceSwap, DiffSwap), face-edit manipulations (FaceEdit), and face-generation manipulations (Image2Image, Text2Image, and DDIM). As shown in Tab. 11, all MDSM-trained models exhibit strong out-of-distribution generalization. Except for ViLT, which performs slightly worse on DiffFace, all models achieve an average accuracy above 65. Our AMD achieves the best performance in several categories, including FaceEdit (54.80), DiffSwap (51.41), Text2Image (59.80), and DDIM (98.38). Overall, AMD attains the highest average accuracy (69.86) on DiffFace, and ranks second on DiFF.

**Scene-level Manipulation.** Beyond facial forgeries, we further evaluate the models on diffusion-based scene-level manipulations, including background modification and full-image generation. We collect 6k additional samples generated via Stable Diffusion v2 Inpainting (Rombach et al., 2022) and Stable Diffusion 3.5 (StabilityAI, 2023), with prompts produced by Qwen2.5-VL (Bai et al., 2025). Results in Tab. 12 demonstrate that MDSM-trained models remain robust even under these entirely novel manipulation types, all achieving average accuracies above 50. Notably, AMD and FKA-Owl both exceed 70 in average accuracy. AMD again outperforms all baselines, achieving the highest overall accuracy (72.10) across scene-level manipulations.

**Face Manipulation.** Our proposed manipulation pipeline is modular and can be easily adapted to integrate alternative manipulation methods. To test this, we replace all face manipulation samples in

Table 12: ACC comparison of MDSM-trained models on scene-level (background) manipulation. The better results in each group are in boldface. AVG refers to the average accuracy across the column.

| Method | Scene-level Manipulation | | AVG |
| --- | --- | --- | --- |
| | Scene Modification | Full-image Generation | |
| HAMMER  (Shao et al., 2023) | 51.29 | 55.76 | 53.53 |
| HAMMER++  (Shao et al., 2024) | 51.38 | 56.03 | 53.70 |
| ViLT  (Kim et al., 2021) | 48.16 | 52.23 | 50.20 |
| FKA-Owl  (Liu et al., 2024) | 85.83 | **57.13** | 71.48 |
| **AMD(Ours)** | **87.96** | 56.23 | **72.10** |

Figure 11: Performance Difference Heatmap between MDSM and $MDSM_{SD}$.

the MDSM test split with those generated by the Stable Diffusion–based methods, DiffSwap  (Zhao et al., 2023) and SD-Face-Editor (Rombach et al., 2022), creating a new test set $MDSM_{SD}$. We then re-evaluate the MDSM-trained models on $MDSM_{SD}$. Results in Tab. 13 confirm that detection accuracies on $MDSM_{SD}$ follow the same trend observed on MDSM (Tab. 2): AMD > FKA-Owl > HAMMER++ > HAMMER. Furthermore, the results in Tab. 11 on unseen diffusion-based manipulations support the same ranking. This consistency demonstrates that models achieving strong performance on MDSM also generalize effectively to diffusion-based manipulations. Additionally, we computed the performance gap between Tab. 13 and Tab. 2, and visualized the performance differences with a heatmap. As shown in Fig. 11, all MDSM-trained models maintain stable test metrics on $MDSM_{SD}$, with the performance drop due to cross-domain effects not exceeding 8 points. Notably, the AMD model trained on MDSM-NYT experienced only a 1.57 percentage point decrease in average ACC on $MDSM_{SD}$. Taken together, these findings confirm that evaluation results on MDSM remain reliable indicators of real-world robustness and our AMD retains strong generalization ability when applied to other types of facial manipulations.

## K  THE VALUE OF INCORPORATING SEMANTIC ALIGNMENT TEXT.

We construct a semantic-aligned dataset with MLLM and its non-aligned counterpart without MLLM for comparison, enabling the quantify the value of incorporating semantic alignment and MLLMs into data construction for training more robust detection models.

In specific, we use the Guardian portion of the MDSM dataset (NA-$MDSM_G$) as a case study. We create its variant termed Non-Alignment $MDSM_G$ (NA-$MDSM_G$) by modifying the text modality in all non-Orig samples. Specifically, for classes involving Text Fabrication manipulations, the original

Table 13: Performance of MDSM-trained models for MDSM$_{SD}$, where the background gray indicates the intra-domain performance. The better results in each group are in boldface. AVG refers to the average performance across five news domains.

| Setting | Method | Test Domain | | | | | | | | | | | | | | | | |
|---|---|---|---|---|---|---|---|---|---|---|---|---|---|---|---|---|---|---|
| | | NYT | | | Guardian | | | USA | | | Wash. | | | BBC | | | AVG | | |
| | | ACC | mAP | mIoU | ACC | mAP | mIoU | ACC | mAP | mIoU | ACC | mAP | mIoU | ACC | mAP | mIoU | ACC | mAP | mIoU |
| Tr. on NYT | ViLT (Kim et al., 2021) | 76.86 | 57.65 | 20.53 | 66.62 | 29.44 | 18.25 | 64.65 | 32.82 | 19.40 | 60.92 | 31.99 | 15.86 | 68.71 | 32.20 | 17.48 | 67.55 | 36.82 | 18.30 |
| | HAMMER (Shao et al., 2023) | 78.17 | 51.22 | 45.89 | 65.88 | 36.80 | 15.72 | 70.59 | 37.75 | 12.99 | 64.19 | 37.96 | 13.01 | 64.18 | 38.19 | 14.86 | 68.60 | 40.38 | 20.50 |
| | HAMMER++ (Shao et al., 2024) | 77.86 | 52.09 | 50.80 | 65.92 | 35.75 | 16.18 | 65.70 | 36.29 | 10.04 | 63.94 | 35.10 | 10.79 | 64.21 | 34.53 | 13.11 | 67.52 | 38.75 | 20.18 |
| | FKA-Owl (Liu et al., 2024) | 92.11 | 73.72 | 54.62 | 77.61 | 40.20 | 41.18 | 75.90 | 42.92 | 46.19 | 76.45 | 47.93 | 41.75 | 83.36 | 57.27 | 40.13 | 81.09 | 52.41 | 44.77 |
| | AMD(Ours) | 90.67 | 77.29 | 67.94 | 79.65 | 59.77 | 58.28 | 77.43 | 64.10 | 48.67 | 82.20 | 65.89 | 56.23 | 83.19 | 59.10 | 65.21 | 82.63 | 65.23 | 59.27 |
| Tr. on Guardian | ViLT (Kim et al., 2021) | 66.94 | 38.11 | 19.65 | 82.19 | 61.80 | 38.75 | 69.45 | 40.75 | 34.02 | 77.49 | 42.59 | 34.22 | 77.84 | 40.14 | 35.19 | 74.78 | 44.68 | 32.37 |
| | HAMMER (Shao et al., 2023) | 58.46 | 33.23 | 19.68 | 75.75 | 47.26 | 49.27 | 71.68 | 47.46 | 40.44 | 73.11 | 47.00 | 46.81 | 76.06 | 48.19 | 52.34 | 71.01 | 44.63 | 41.71 |
| | HAMMER++ (Shao et al., 2024) | 58.18 | 35.89 | 22.03 | 76.22 | 54.11 | 60.23 | 71.29 | 44.23 | 40.11 | 73.97 | 47.16 | 41.87 | 77.17 | 48.23 | 53.84 | 71.37 | 45.92 | 43.61 |
| | FKA-Owl (Liu et al., 2024) | 79.77 | 36.92 | 25.04 | 90.02 | 71.36 | 68.87 | 79.95 | 48.74 | 46.50 | 76.98 | 48.19 | 45.85 | 86.51 | 62.60 | 57.65 | 82.65 | 53.56 | 48.78 |
| | AMD (Ours) | 82.29 | 44.90 | 49.10 | 89.03 | 75.01 | 81.23 | 84.23 | 48.19 | 48.82 | 86.93 | 50.07 | 47.50 | 87.19 | 59.36 | 58.13 | 85.94 | 55.51 | 56.96 |

Table 14: Comparison on NA-MDSM$_G$ and NA-MDSM$_G$

| Test Domain | Train Domain | | | | | |
|---|---|---|---|---|---|---|
| | MDSM$_G$ | | | NA-MDSM$_G$ | | |
| | ACC | mAP | mIoU | ACC | mAP | mIoU |
| MDSM$_G$ | 91.43 | 80.85 | 85.09 | 72.31 | 56.93 | 65.07 |
| NA-MDSM$_G$ | 88.57 | 75.63 | 79.31 | 94.14 | 84.26 | 85.79 |
| AVG | 90.00 | 78.24 | 82.20 | 83.22 | 70.60 | 75.43 |
| **Performance Drop** | ↓ 2.86 | ↓ 5.22 | ↓ 5.78 | ↓ 21.82 | ↓ 27.33 | ↓ 20.72 |

caption is replaced with a randomly sampled caption from a pool of real news articles (excluding its own). For all other manipulated classes, we retain the original caption.

AMD is trained on both sets. As shown in Tab. 14, MDSM$_G$ yields better average performance and generalization. AMD trained on MDSM$_G$ transfers well to NA-MDSM$_G$ (ACC: 91.43→88.57, 96.86% retained), confirming that semantic alignment supports generalization to traditional mismatches. In contrast, AMD trained on NA-MDSM$_G$ degrades sharply on MDSM$_G$ (ACC/mAP/mIoU drop by 21.83/27.33/20.72), showing AMD trained solely on traditional mismatches struggles with finer-grained cross-modal aligned manipulations.

## L    ARTIFACT TOKEN VISUALIZATION

To elucidate the operational dynamics of artifact tokens within our AMD framework, we employ t-distributed stochastic neighbor embedding (t-SNE) (van der Maaten & Hinton, 2008) to project the high-dimensional token representations into a low-dimensional space. Visualization is performed at both the sequence scale and the individual token scale to reveal how discriminative and redundant information evolves throughout the AMD pipeline.

### L.1    SEQUENCE-SCALE VISUALIZATION

In the Fig.5 of the main paper, we randomly sample 2,400 examples from the MDSM test set. For each example, the portion of the token sequence corresponding to artifact tokens is aggregated into a single representation and then reduced to two dimensions via t-SNE. We visualize three key processing stages:

- **Pre-encoding (raw input)**: Artifact tokens before any encoding, $E_a$. (Fig.5a)
- **Post-pre-perception encoding**: Tokens after passing through the Artifact Pre-perception Encoder stage, $\hat{E}_a$. (Fig.5b)

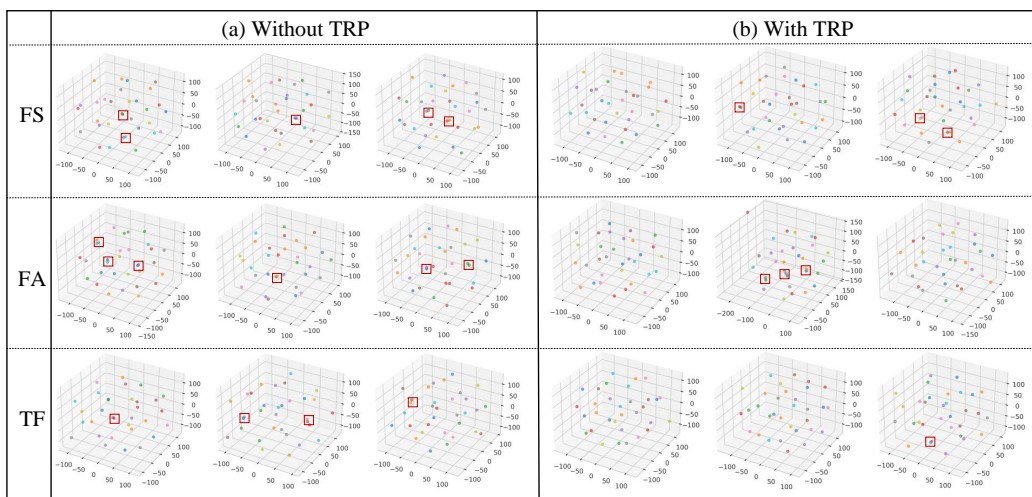

Figure 12: Visualization of $\hat{E}_a^m$ tokens in $S_m$. TRP, FS, FA, TF indicates Token Redundancy Penalty, Face Swap, Face Attribute, and Text Fabrication, respectively.

- **Post-manipulation reasoning**: Tokens after the encoder process of Manipulation-Oriented Reasoning stage, $\hat{E}_a^m$. (Fig.5c)

Prior to any encoding, the t-SNE embedding exhibits a highly intermixed distribution, with no obvious separation between original samples (blue) and manipulated samples. After pre-perception encoding, original samples form a distinct cluster, indicating that $\mathcal{E}_m^p$ accumulates coarse-grained knowledge for true versus false discrimination. Finally, following the manipulation-oriented reasoning stage, original samples become more tightly clustered and manipulated samples (of various types) arrange into more coherent subclusters, demonstrating that $\hat{E}_a^m$ encodes finer-grained information about manipulation categories.

### L.2 TOKEN-SCALE VISUALIZATION

In our AMD framework, each artifact token consists of 32 sub-token embeddings. To mitigate the randomness associated with small sample sizes, we aggregate 100 samples into a single group: for each group, we first compute the mean of the 100 corresponding artifact token sequences to obtain one representative sequence, and then apply t-SNE to project its 32 sub-token embeddings into a three-dimensional space. For each of the three manipulation classes (Face Swap, Face Attribute, Text Fabrication) we form three independent groups, yielding a total of nine 3D visualizations. These visualizations are generated under two experimental conditions:

- **Without Token Redundancy Penalty (TRP)** (Fig.12a).
- **With TRP** (Fig.12b).

We approximately assume that sub-token embeddings which overlap in the 3D t-SNE plot carry highly similar information and may represent redundancy. In the absence of TRP, Fig.12a, we observe 15 instances of such overlapping sub-token points (highlighted by red boxes), indicating that many sub-tokens are encoding near-duplicate features. while introducing TRP, as shown in Fig.12b, the number of overlaps decreases to 7, demonstrating that the penalty encourages each sub-token to capture more distinct and complementary information. This increase in token diversity is positively correlated with improved manipulation detection performance.

Through sequence-scale and token-scale t-SNE visualizations, we demonstrate that 1) the Pre-perception Encoder progressively separates genuine from manipulated samples, 2) the Manipulation-Oriented Reasoning refines class-specific features, and 3) the Token Redundancy Penalty effectively increases sub-token diversity within artifact tokens, thereby strengthening AMD's discriminative power.

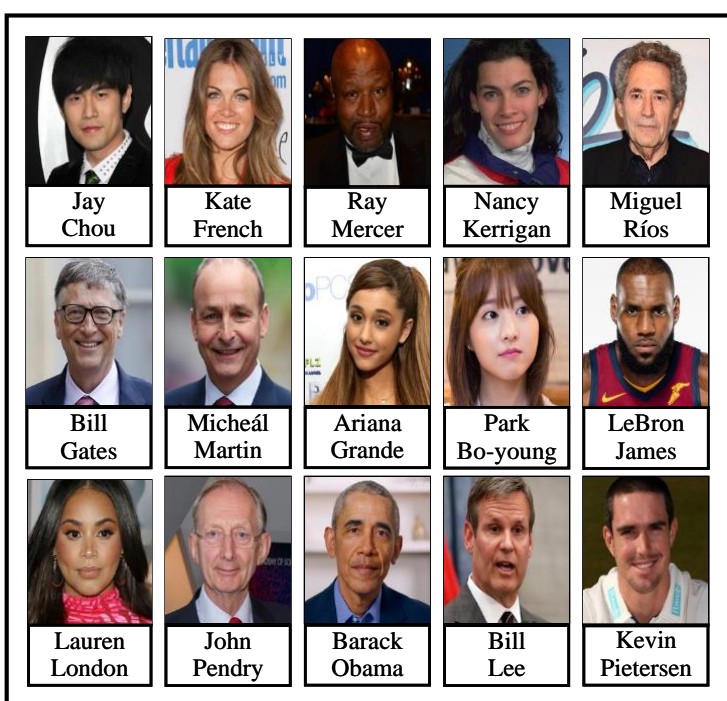

Figure 13: Examples of Celebrity Head-shot Dataset.

# M    CELEBRITY HEAD-SHOT DATASET

While some celebrity datasets  (Tero et al., 2018; Lee et al., 2020; Liu et al., 2015) have been created, they typically do not provide a comprehensive mapping between individual names and their corresponding head-shots. To advance the research problem in this paper, we construct the **Celebrity Head-shot Dataset**. Using the celebrity directory provided by the Pantheon  (World, 2025), we select names of celebrities from fields with significant public influence, such as politics, religion, and diplomacy, focusing exclusively on those who are still alive. For each name, multiple images were collected, and after the scraping process, we use the MLLM Qwen2-VL  (Wang et al., 2024) to filter and select the highest-quality image as the final result. After filtering and compiling the data, the Celebrity Head-shot Dataset contains a total of 29,697 pairs of names and head-shots, some examples are shown in Fig. 13.

# N    CASE ANALYSIS

Fig. 14 illustrates the performance of AMD and other comparative models on the MDSM test set. In Case 1, all tested models correctly identified the manipulation type of test samples. However, it is noteworthy that although ViLT, HAMMER, and FKA-Owl correctly classify the manipulation type, they still generate grounding boxes.  This issue also appears in other samples.  In Case 2, modal-alignment-based methods, ViLT and HAMMER, fail to correctly determine the manipulation type.  For instance, in Fig. 14(d), the manipulation detail involves replacing Obama's face with Revolori's and subsequently generating a semantically coherent narrative by MLLM. This confuses modal-alignment-based methods, leading to misclassification.  In Case 3, only AMD correctly identify all samples.  Notably, in Fig. 14(h), the forgery knowledge-augmented FKA-Owl model detects image modality manipulation and generates a detection box pointing to the glass reflection. HAMMER exhibits the same issue. These visualized cases further demonstrate the effectiveness and superiority of our proposed AMD model, establishing it as a unified solution for combating advanced MLLM-driven deception.

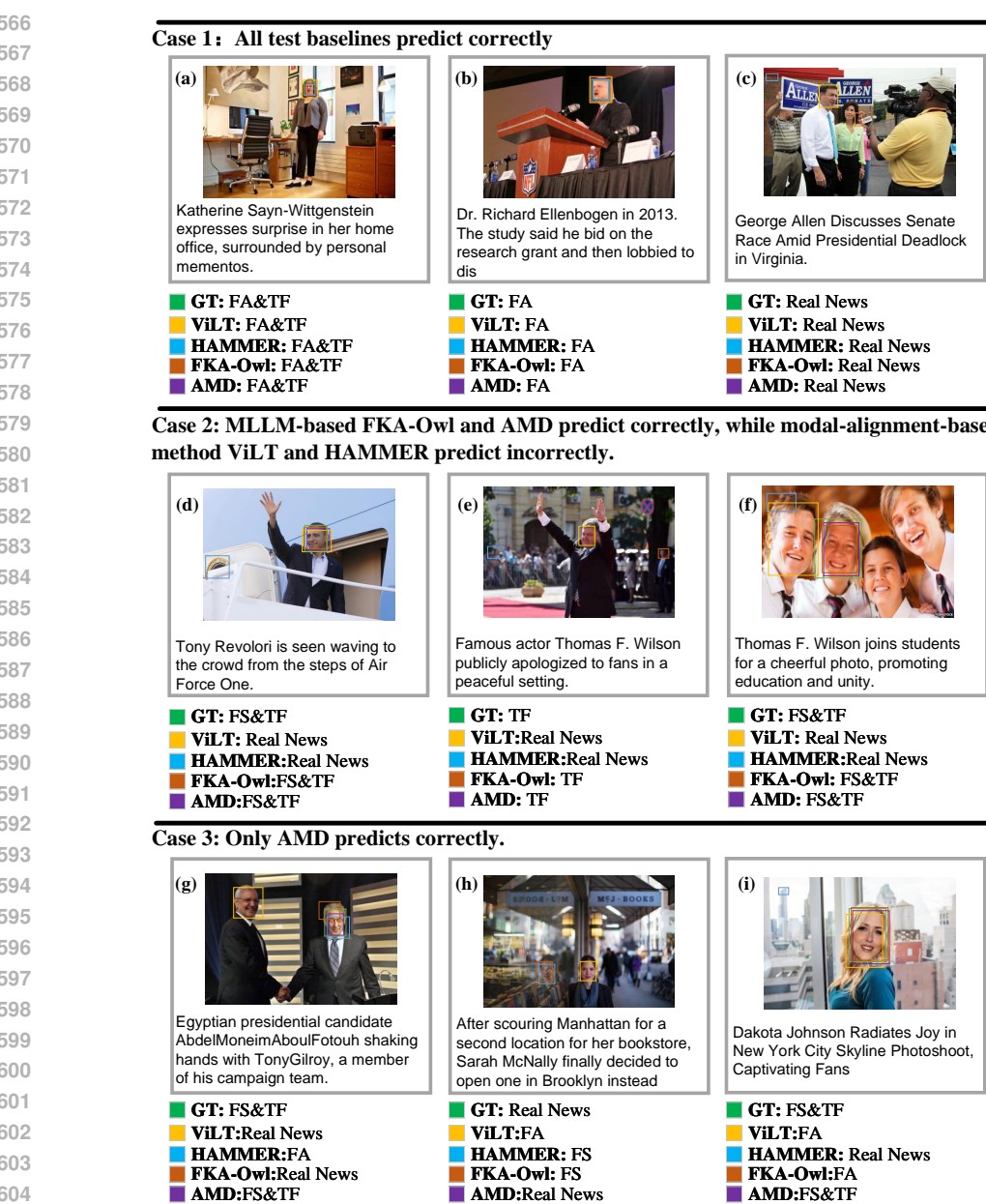

Figure 14: Cases in the testing set. FS, FA, TF indicates Face Swap, Face Attribute, and Text Fabrication, respectively.

## O    LARGE LANGUAGE MODEL USAGE STATEMENT

This paper used large language models solely for text polishing and expression refinement. No large language models were involved in other aspects of the research, including data collection, experimental design, result analysis, or conclusion derivation.

