# OpenReview forum: "The Coherence Trap: When MLLM-Crafted Narratives Exploit Manipulated Visual Contexts"
_ICLR.cc/2026/Conference — ICLR 2026 Conference Withdrawn Submission_

### Official Review · Reviewer_QRQG · 2025-10-31

**Soundness:** 3
**Presentation:** 3
**Contribution:** 3
**Rating:** 6
**Confidence:** 4

**Summary:**

This paper looks at investigating how MLLM driven misinformation in two modalities, namely text and image. They construct a dataset MDSM that simulates realistic multimodal manipulations where images are first manipulated and paired with MLLM-generated deceptive texts that maintain semantic consistency with the visual manipulations. Additionally this paper proposes a detection and grounding framework that outputs both coordinates of the manipulation and explanations (AMD).

**Strengths:**

* I believe this paper is relevant and has a few strengths to it. There are some interesting experiments that are done with both a zero-shot setting and training models on the MDSM dataset.

* Authors spent time trying to do LoRA finetuning on their dataset which I think was an important experiment the have included.

* Showcasing how other models like HAMMER and FKA-Owl which are prevalent in multi-modal manipulations was good to have and some discussion of the analysis

* Including a human evaluation of some of the manipulated images/text was good to include for this work to showcase how humans can be fooled by multi-modal models

**Weaknesses:**

* I believe that the authors should try and include more Open-Source multimodal models, for zero-shot evaluation in Table 2, currently the only model present is Qwen and no other popular models like Deepseek, LLaVa, Yi-VL.

[1] Deepseek llm: Scaling open-source language models with longtermism.
[2] Visual instruction tuning, Neurips 2023
[3] Yi: Open foundation models by 01.ai, 2024.

**Questions:**

* Did the authors use GPT-4o and Gemini-2.0 on all the images for the test set? Seems like quite a large test set for a paid model. How large is the test set?

* Why didn’t the authors include more Open-Source models in Table 2, currently there is basically one 1, since only variants of QWEN is included.

---

> ### Author Response · Authors · 2025-11-13
> **Response to Reviewer QRQG**
>
> Thank you for the constructive comments, we provide the following clarifications:
> ## 4-1 Regarding including more multimodal models for zero-shot evaluation.
>
> Thank you for your suggestion. We have included additional open-source multimodal models for zero-shot evaluation to ensure a more comprehensive comparison, Including LLaVa1.6[a], DeepSeek-VL2[b] and Yi-VL[c]. The updated results are shown in Table X4-1, which reports the zero-shot performance of representative general-purpose multimodal models on our MDSM dataset.
>
> **Table X4-1. Zero-shot evaluation results of representative general-purpose multimodal models on the MDSM dataset.**
>
> ****
>
> | Method           | $NYT_{ACC}$ | $NYT_{mAP}$ | $NYT_{mIoU}$ | $Guardian_{ACC}$ | $Guardian_{mAP}$ | $Guardian_{mIoU}$ | $USA_{ACC}$ | $USA_{mAP}$ | $USA_{mIoU}$ | $Wash_{ACC}$ | $Wash_{mAP}$ | $Wash_{mIoU}$ | $BBC_{ACC}$ | $BBC_{mAP}$ | $BBC_{mIoU}$ | $AVG_{ACC}$ | $AVG_{mAP}$ | $AVG_{mIoU}$ |
> | ---------------- | ----------- | ----------- | ------------ | ---------------- | ---------------- | ----------------- | ----------- | ----------- | ------------ | ------------ | ------------ | ------------- | ----------- | ----------- | ------------ | ----------- | ----------- | ------------ |
> | LLaVa-v1.6-13B |32.10 | 21.86 | 0.00 | 21.76 | 19.18 | 0.00 | 17.80 | 12.76 | 0.00 | 15.76 | 25.47 | 0.00 | 22.70 | 17.38 | 0.00 | 22.02 | 19.33 | 0.00 |
> | DeepSeek-VL2-27B | 38.12       | 21.79       | 0.45         | 30.35            | 20.12            | 0.76              | 22.49       | 37.27       | 0.43         | 21.73        | 20.02        | 1.78          | 29.70       | 26.58       | 1.12         | 28.48       | 25.16       | 0.91         |
> | Yi-VL-34B        | 45.74       | 24.19       | 0.07         | 34.98            | 23.98            | 0.13              | 20.63       | 36.74       | 0.00         | 19.28        | 19.10        | 0.00          | 31.39       | 26.97       | 0.00         | 30.40       | 26.20       | 0.04         |
> | Qwen2.5-VL-72B   | 47.74       | 29.24       | 0.00         | 35.18            | 25.70            | 0.00              | 24.66       | **40.60**   | 0.00         | 25.11        | 40.29        | 0.28          | 35.89       | 31.51       | 0.00         | 33.72       | 33.47       | 0.06         |
> | Qwen3-VL-235B    | 45.29       | 25.01       | 0.71         | 38.12            | 27.41            | 1.19              | 22.87       | 39.17       | 0.10         | 22.87        | **40.77**    | 1.34          | 37.22       | **31.60**   | 0.98         | 33.27       | **33.69**   | 0.86         |
> | GPT-4o           | 48.48       | 27.90       | 0.82         | 35.68            | **29.49**        | **1.23**          | 24.62       | 39.88       | 1.37         | 23.62        | 38.89        | 1.22          | 37.19       | 30.48       | 1.20         | 33.92       | 33.33       | 1.17         |
> | Gemini-2.0       | **56.05**   | **33.16**   | **1.44**     | **41.26**        | 24.37            | 1.12              | **29.60**   | 38.29       | **1.40**     | **29.15**    | 35.20        | **2.42**      | **38.12**   | 29.13       | **2.25**     | **38.83**   | 32.03       | **1.72**     |
>
>
>
> ## 4-2 Did the authors use GPT-4o and Gemini-2.0 on all the images for the test set?
>
> Yes, GPT-4o and Gemini-2.0 are tested on all 440,73 test samples. They are both paid model and costs around 292 US dollars for GPT-4o and 148 US dollars for Gemini-2.0.
>
>
> [a] Liu et.al. Visual instruction tuning. NeurIPS 2023.
>
> [b] Wu et.al. Deepseek-vl2: Mixture-of-experts vision-language models for advanced multimodal understanding. arXiv 2024.
>
> [c] 01. AI. Yi: Open Foundation Models by 01.AI. arXiv 2024.

---

### Official Review · Reviewer_GjoA · 2025-10-31

**Soundness:** 2
**Presentation:** 2
**Contribution:** 2
**Rating:** 4
**Confidence:** 4

**Summary:**

This paper focuses on the "coherence trap" phenomenon in multimodal misinformation detection. The authors first construct MDSM, a large-scale, diverse, and aligned benchmark for multimodal manipulation detection, comprising challenging samples from reputable news sources. To address this challenge, the authors propose AMD (Artifact-aware Manipulation Diagnosis), a novel framework built on Florence-2 that integrates artifact pre-perception encoding and manipulation-oriented reasoning. Experiments show that AMD outperforms existing methods on MDSM, demonstrating improved robustness in detecting coherently generated fake news.

**Strengths:**

1. The paper identifies and formalizes the "coherence trap," a highly relevant and critical issue in the era of advanced generative models, where the very coherence that makes AI-generated content useful also makes it dangerously deceptive.

2. The construction of MDSM is a significant contribution. Its scale, diversity of sources, and alignment between modalities make it a valuable resource for the research community.

3. The paper presents thorough experiments, including ablation studies and cross-domain evaluations, demonstrating the superior performance of AMD over state-of-the-art baselines on the MDSM benchmark.

**Weaknesses:**

1. While the paper formally defines and highlights the "coherence trap" as a critical challenge in multimodal misinformation detection, the underlying concept of detecting semantically aligned fake content is not entirely novel. Prior works, such as MMFakeBench, have already explored scenarios involving coherent image-text manipulations.

2. The proposed AMD framework is built upon the powerful Florence-2 model and leverages its strong pre-trained multimodal understanding. The architectural innovation of AMD itself appears limited.

3. While the paper shows AMD's overall success, a deeper analysis of when and why AMD fails (e.g., specific types of manipulations it struggles with, examples of false positives/negatives) would strengthen the work and provide more insight for future research.

[1]Liu X, Li Z, Li P P, et al. MMFakeBench: A Mixed-Source Multimodal Misinformation Detection Benchmark for LVLMs. ICLR 2024

**Questions:**

1. While the "coherence trap" is well-motivated, similar aligned fakes have been studied in prior work (e.g., MMFakeBench). How does this work differ in problem formulation or threat model beyond dataset scale?

2. The AMD framework builds directly on Florence-2 with minimal architectural changes. To what extent do the gains come from the model design versus the strong pre-trained backbone?

3. What are the main failure modes of AMD? A deeper analysis of false positives/negatives or challenging manipulation types (e.g., subtle edits, coherent text-only fakes) would strengthen the paper.

4. How generalizable is AMD beyond news domains? The reliance on Florence-2’s world knowledge may limit performance on out-of-distribution content (more sophisticated, MLLM-generated manipulations or content generated by different large models).

---

> ### Author Response · Authors · 2025-11-13
> **Response to Reviewer GjoA**
>
> Thank you for the detailed comments, we provide the following clarifications:
> ## 3-1 Discrepancy between our MDSM and MMFakeBench.
> We sincerely thank you for your insightful comments. However, there are several key differences between MMFakeBench and MDSM:
> (1) As the dataset name suggests, MDSM primarily focuses on MLLM-crafted misinformation, which is not present in MMFakeBench.
>
> (2 ) MMFakeBench contains only 1.1k samples and serves solely as an evaluation benchmark, whereas MDSM includes 441k samples and provides a large-scale standard dataset for both model training and evaluation.
>
> (3) All samples in MDSM are 100% semantically aligned, while MMFakeBench contains only about 30% semantically aligned samples.
>
> ## 3-2 Regarding the architectural innovation of AMD.
>
> It is a great point. We did not make significant architectural changes to Florence 2; instead, we applied appropriate constraints from auxiliary tasks to enhance the performance. This approach offers two key benefits:
>
> (1) Efficiency: The auxiliary modules are discarded during inference, resulting in no additional computational burden, as shown in Table 7d (line 1148).
>
> (2) Simplicity: Since no special/complex architectural modifications are introduced, our network is almost identical to Florence 2 at inference time. In contrast, FKA-Owl incorporates multiple architectural changes, leading to a heavier model and slower inference (Table 7d). We believe that promoting simpler architectures is preferable to relying on complex designs.
>
> ## 3-3 What are the main failure modes of AMD?
>
> In our practice, the *FaceAttribute&TextFabrication* type represents the most challenging cases. We will include a clearer case analysis in the revised version of the paper.
>
> ## 3-4 How generalizable is AMD beyond news domains?
> Please refer to Tables 11 (line 1316) and 12 (line 1350), which demonstrate the generalization of our AMD framework to a wider range of image types beyond news samples, while Table 4(c) reports performance across different LLMs. These results collectively validate the strong generalization capability of our approach.

---

### Official Review · Reviewer_6UUQ · 2025-11-01

**Soundness:** 3
**Presentation:** 2
**Contribution:** 2
**Rating:** 4
**Confidence:** 5

**Summary:**

The paper tackles multimodal misinformation detection under the claimed more realistic threat model, generating images that have been locally manipulated (via face swap or face attribute editing), and paired with text generated by a multimodal LLM (MLLM) that is semantically consistent with the manipulation. The authors propose MDSM, a 441k-sample dataset where faces in news images are edited and paired with coherent deceptive narratives. Then introduce AMD, a Florence-2–based model that can handle both (i) binary “fake vs real,” (ii) manipulation type, and (iii) manipulated-region bounding boxes tasks in text form.

**Strengths:**

1. The motivation is clear. The paper explicitly argues that prior work assumes crude cross-modal inconsistency, which makes detection too easy because the text and image obviously disagree. By contrast, MDSM uses an MLLM to generate fluent, contextually aligned fake narratives that match the manipulated visual identity. This is a meaningful direction.

2. AMD outputs manipulation decisions and the tampered region coordinates as a single textual answer instead of separate detection heads. This makes the downstream application easier.

3. They employ a cross-domain setting for evaluation, emphasizing the generalization ability.

**Weaknesses:**

1. The novelty claimed based on the flaw of previous works is not strong enough. Prior work like FKA-Owl is already an MLLM-style system that “incorporates more world knowledge to improve the model’s cross-domain performance,” explicitly targeting multimodal fake news scenarios. The paper acknowledges this but still claims current approaches “fail to account for sophisticated misinformation synthesized by MLLMs,” which is not persuasive enough.

2. On the model side, AMD is essentially Florence-2 + DaViT with several augmented function modules. This is solid engineering, but conceptually similar to known frameworks in this area.

3. The paper repeatedly stresses that MDSM “defines a more challenging and practical problem,” “high-risk disinformation,” and “semantically coherent and contextually plausible narratives,” and that previous benchmarks are “too simplistic to effectively deceive the public.” Are there any quantitative and empirical demonstrations of the claimed risk?

4. MDSM only keeps “human-centric” news with faces and named entities, then applies two manipulation types in the image domain, including Face Swap and Face Attribute editing. This is not a general-purpose “multimodal misinformation” benchmark. A multimodal celebrity face tampering benchmark is more appropriate.

5. In the cross-domain training setting, how could the data leakage risk be avoided? Would there be the same celebrity headshot in training and testing sets? Besides, the distribution of manipulations and the linguistic style of fabricated text are almost identical across “domains.” It looks closer to the same attacker pipeline, different news source name.

6. In the tables, you compare single mAP values across models: how exactly is multi-label type detection scored as AP? Is it per manipulation type and averaged? Macro/micro? The paper does not say.

7. Which parameters are actually trainable in APE? Only the Artifact Token + classifier Ca? Is there a two-stage schedule or joint multitask training with loss L? The description is not clear enough for training details.

8. In ablations, why do the authors never isolate TRP alone?

9. If the AMD decoder is effectively trained to emit the answer at the end of that rigid QA format, how robust is it to prompt perturbations? Could an adversary just reformulate the question?

**Questions:**

Please see the question above.

---

> ### Author Response · Authors · 2025-11-13
> **Response to Reviewer 6UUQ**
>
> Thank you for the detailed comments, we provide the following clarifications:
> ## 2-1 Discrepancy with FKA-OWL and our novelty.
> We fully understand your concerns, but there are many differences between FKA-Owl and our work:
> (1) FKA-OWL is trained on the DGM4 dataset, whose text manipulations follow fixed rules instead of being synthesized by MLLMs. Therefore, it cannot effectively detect MLLM-generated misinformation, which is the novel focus of our work. Table 2 further validates this, showing FKA-OWL’s inferior mAP on fine-grained forgery detection.
>
> (2) While FKA-OWL also employs an MLLM backbone, it only performs binary real/fake detection and does not support fine-grained forgery classification or region grounding.
>
> (3) Our novelty primarily lies in three aspects: (a) the exploration of a new threat scenario introduced by MLLMs; (b) the introduction of a learnable artifact token that collaborates with artifact pre-perception encoding and manipulation-oriented reasoning modules to effectively capture forgery cues; and (c) the design of a token redundancy penalty constraint that encourages the learnable token to aggregate diverse forgery-related evidence.
>
> ## 2-2 Regarding the model design.
> It is a great point. We respectfully clarify that taking an MLLM as the backbone for multimodal manipulation is neither new nor one of our claimed contributions. In fact, directly applying an off-the-shelf MLLM to our task yields suboptimal performance. As shown in Table 4(a), simply fine-tuning Florence achieves only 76% accuracy on the NYT domain.To address this limitation, we introduce several new designs to optimize performance:
>
> (1) a learnable artifact token, which collaborates with artifact pre-perception encoding and manipulation-oriented reasoning to effectively capture forgery information;
>
> and (2) a token redundancy penalty constraint, which encourages the learnable token to aggregate diverse forgery cues.
>
> These components are new and have not been explored in prior works. Moreover, unlike previous methods that rely on multiple task-specific heads for manipulation detection and grounding, our AMD framework unifies these tasks into a single end-to-end architecture, enabling both detection and grounding within one prediction module.
>
> ## 2-3 Are there any quantitative and empirical demonstrations of the claimed risk?
> Yes, please refer to Table Figure 7 (line 882) and Table 14 (line 1419), which show that our dataset is more challenging. Human evaluators reach only 56.7% on MDSM (Figure 7c) versus 63.4% on DGM4. Table 14 shows that a model trained on the semantically mismatched $\text{NA-MDSM}_G$ drops by 21.82, 27.33, and 20.72 points in ACC, mAP, and mIoU when tested on MDSM, indicating difficulty in recognizing semantically coherent traps.
> ## 2-4  A multimodal celebrity face tampering benchmark is more appropriate.
> Thank you for the suggestion. As stated in our paper (line 145, page 3), MDSM is a human-centric dataset. Following DGM4, which is also human-centric, our dataset name does not explicitly indicate this aspect; instead, it emphasizes the core feature of MLLM-induced threats.
> ## 2-5  How could the data leakage risk be avoided in cross-domain training set?
> Our evaluation follows FKA-Owl, treating different news sources as separate domains for generalization testing. Due to the dataset’s large scale, verifying repeated samples or headshots across domains is difficult. Nevertheless, Table 2 shows that models trained on one domain suffer substantial performance drops on others—for example, FKA-Owl achieves 94.6% on NYT but only 77.2% on Guardian and 78.0% on USA—indicating significant domain shifts. Thus, data leakage is unlikely, and our cross-domain evaluation remains reliable.
> ## 2-6 Regarding the mAP in Tables.
> For fairness of comparison and reproducibility, we follow the DGM4 [1] evaluation protocol, which computes the Average Precision (AP) for each manipulation type independently and reports the mean across all types (macro mAP). Specifically, the mAP values reported in all tables are obtained by averaging the per-class APs over all manipulation types:
> $ \text{mAP} = \frac{1}{K} \sum_{k=1}^{K} \text{AP}_k. $
>
> Therefore, the reported metric corresponds to a **macro-level mAP**, rather than a micro-averaged one.
> [1] Shao et.al. Detecting and Grounding Multi-Modal Media Manipulation and Beyond. TPAMI, 2024.
> ## 2-7 Which parameters are actually trainable in APE?
> The Artifact Token, Classifier Head, and Attention Pooling module are jointly trained in a single optimization stage.
> ## 2-8  Why do the authors never isolate TRP alone?
> The pure contribution of TRP is clearly validated in the last row of Table 4 (a).
> ## 2-9 Regarding the attack of prompt perturbations?
> We fully understand your concerns. However, it is worth noting that attacks via prompt perturbations are not considered in this work. We plan to explore this aspect in future research.

---

### Official Review · Reviewer_3weR · 2025-11-01

**Soundness:** 2
**Presentation:** 3
**Contribution:** 2
**Rating:** 2
**Confidence:** 2

**Summary:**

This paper introduces a novel forgery detection framework that targets semantically coherent multimodal content generated by modern Multimodal Large Language Models (MLLMs). The proposed framework leverages Artifact Pre-perception Encoding (APE) and Manipulation-Oriented Reasoning (MOR) to collaboratively analyze image-text manipulations through the reasoning capability of MLLMs. In addition, the authors construct an MLLM-Driven Synthetic Multimodal (MDSM) dataset, where image and text modifications are jointly guided by MLLMs to ensure semantic alignment.

**Strengths:**

1.The paper clearly highlights the risks posed by semantically coherent forgeries generated by modern Multimodal Large Language Models (MLLMs).
2.It presents a large-scale, semantically aligned multimodal dataset, effectively filling a crucial gap in resources for studying MLLM-driven misinformation.
3.The proposed framework integrates Artifact Pre-perception Encoding (APE) and Manipulation-Oriented Reasoning (MOR), leveraging the reasoning capabilities of MLLMs to collaboratively analyze image-text manipulations. By synergizing APE and MOR, it effectively adapts MLLMs for precise manipulation analysis.
4.The authors conduct comprehensive experiments on both the MDSM and DGM4 datasets, achieving state-of-the-art average cross-domain generalization performance.

**Weaknesses:**

1.The paper aims to tackle the challenge of forgery detection in semantically aligned image-text scenarios. However, its model design primarily focuses on visual forgeries while largely overlooking textual forgeries.
2.Limited interpretability: although some visualizations are provided, the paper does not clearly demonstrate which specific forgery cues the model captures, nor does it offer a human-understandable reasoning process behind its decisions.
3.The organization of Section 2.2 could be improved — the relationships among its subsections are not clearly articulated, making the overall flow somewhat difficult to follow.

**Questions:**

1.Could the authors elaborate on the choice to focus specifically on facial modifications and replacements? Have considerations been made regarding the potential extension of the framework to other types of visual or contextual manipulations, such as scene-level or object-level edits?
2.Would the authors be able to provide additional details about the dataset used to train the artifact-aware classification head? In particular, it would be helpful to know the scale of the data and the relative proportions of different manipulation types included during training.

---

> ### Author Response · Authors · 2025-11-13
> **Response to Reviewer 3weR**
>
> Thank you for the detailed comments, we provide the following clarifications：
>
> ## 1-1 However, its model design primarily focuses on visual forgeries while largely overlooking textual forgeries？
> It is a great point. All modules in our framework, except for the Grounding loss and the TPR constraint, jointly consider both visual and textual features. Textual forgeries are explicitly modeled within these modules.
>
> ## 1-2 Limited interpretability.
> We fully understand your concerns and we acknowledge the importance of interpretability as a research direction. But in our work concentrates on enhancing detection performance for MLLM-crafted misinformation, rather than pursuing reasoning or interpretability aspects.
>
> ## 1-3 The organization of Section 2.2 could be improved — the relationships among its subsections are not clearly articulated, making the overall flow somewhat difficult to follow.
>  We sincerely thank you for the detailed review. Section 2.2 outlines how images and text are manipulated. We first describe image manipulations, followed by text manipulations designed to maintain semantic consistency. We believe that the structure of this section reflects a logical and clear workflow. We will pay special attention to this part and further refine its presentation in the revised manuscript.
>
> ## 1-4 Could the authors elaborate on the choice to focus specifically on facial modifications and replacements?
> It is a great point. The reason we focus on facial modifications mainly stems from three aspects: (1) celebrity-related news spreads rapidly on social media and thus poses higher misinformation risks; (2) celebrity-related news constitutes a large proportion and represents one of the most influential categories in social media platforms; and (3) the rapid progress in face manipulation techniques makes celebrity faces particularly easy to modify. Consequently, many existing studies, such as DMG4 and ours, naturally focus on celebrity-related news.
>
> ## 1-5 Have considerations been made regarding the potential extension of the framework to other types of visual or contextual manipulations, such as scene-level or object-level edits?
> Yes, in our future work, we plan to extend the current MSDM to include additional types of visual edits as mentioned by the reviewer.
>
> ## 1-6  Details of dataset used to train the artifact-aware classification head.
> All training samples (~121k text–image pairs in the Guardian-training setting and ~140k pairs in the NYT-training setting) are used to train the artifact-aware classification head. The distribution of samples across different manipulation types is presented in **Fig. 3(a)** , please refer toline 181 in the paper.

---

### Note · Authors · 2025-11-14

I have read and agree with the venue's withdrawal policy on behalf of myself and my co-authors.